# Cell-autonomous requirement for ACE2 across organs in lethal mouse SARS-CoV-2 infection

**Alan T. Tang[1]☼, David W. Buchholz[2]☼, Katherine M. Szigety[1], Brian Imbiakha[2], Siqi Gao[1], Maxwell Frankfurter[1], Min Wang[1], Jisheng Yang[1], Peter Hewins[3], Patricia Mericko-Ishizuka[1], N Adrian Leu[4], Stephanie Sterling[4], Isaac A. Monreal[2], Julie Sahler[2], Avery August[2], Xuming Zhu[5,6], Kellie A. Jurado[3], Mingang Xu[5,6], Edward E. Morrisey[1,7], Sarah E. Millar[5,6,8], Hector C. Aguilar[2]*, Mark L. Kahn[1]***

**1** Department of Medicine and Cardiovascular Institute, University of Pennsylvania, Philadelphia, Pennsylvania, United States of America, **2** Department of Microbiology and Immunology, College of Veterinary Medicine, Cornell University, Ithaca, New York, United States of America, **3** Department of Microbiology, University of Pennsylvania Perelman School of Medicine, Philadelphia, Pennsylvania, United States of America, **4** Department of Biomedical Sciences, School of Veterinary Medicine, University of Pennsylvania, Philadelphia, United States of America, **5** Black Family Stem Cell Institute, Icahn School of Medicine at Mount Sinai, New York, New York, United States of America, **6** Department of Cell, Developmental and Regenerative Biology, Icahn School of Medicine at Mount Sinai, New York, New York, United States of America, **7** Penn-CHOP Lung Biology Institute, Perelman School of Medicine, University of Pennsylvania, Philadelphia, Pennsylvania, United States of America, **8** Department of Dermatology, Icahn School of Medicine at Mount Sinai, New York, New York, United States of America

☼ These authors contributed equally to this work.
* ha363@cornell.edu (HCA); markkahn@pennmedicine.upenn.edu (MLK)

**Data Availability Statement:** All the data and reagents are provided in the manuscript and supporting files, or available commercially, except for the new transgenic mouse lines (hACE2fl,

## Abstract

Angiotensin-converting enzyme 2 (ACE2) is the cell-surface receptor for Severe Acute Respiratory Syndrome Coronavirus 2 (SARS-CoV-2). While its central role in Coronavirus Disease 2019 (COVID-19) pathogenesis is indisputable, there remains significant debate regarding the role of this transmembrane carboxypeptidase in the disease course. These include the role of soluble versus membrane-bound ACE2, as well as ACE2-independent mechanisms that may contribute to viral spread. Testing these roles requires in vivo models. Here, we report humanized ACE2-floxed mice in which hACE2 is expressed from the mouse *Ace2* locus in a manner that confers lethal disease and permits cell-specific, Cre-mediated loss of function, and LSL-hACE2 mice in which hACE2 is expressed from the *Rosa26* locus enabling cell-specific, Cre-mediated gain of function. Following exposure to SARS-CoV-2, hACE2-floxed mice experienced lethal cachexia, pulmonary infiltrates, intra-vascular thrombosis and hypoxemia—hallmarks of severe COVID-19. Cre-mediated loss and gain of hACE2 demonstrate that neuronal infection confers lethal cachexia, hypoxemia, and respiratory failure in the absence of lung epithelial infection. In this series of genetic experiments, we demonstrate that ACE2 is absolutely and cell-autonomously required for SARS-CoV-2 infection in the olfactory epithelium, brain, and lung across diverse cell types. Therapies inhibiting or blocking ACE2 at these different sites are likely to be an effective strategy towards preventing severe COVID-19.

hACE2hypo, and R26-LSL-hACE2). These mouse lines are available through a material transfer agreement with the University of Pennsylvania; interested researchers should contact the Principal Investigator, Mark Kahn, at markkahn@pennmedicine.upenn.edu and CC the Office of Research Services at ORSMTA@pobox. upenn.edu. The remaining mouse lines used in this manucript are available at public repositories.

**Funding:** This work was supported by National Institute of Health grants R01HL39552-04S1 (MK), AHA 963048 (MK), R01AI109022 and R21AI156731 (HAC), T32EB023860 (DWB), R25GM125597 (BI), and R01AI138370 (BI and AA), an AHA Postdoctoral Fellowship #906488 (SG) and a Penn CVI Dream Team grant (MK). The funders had no role in study design, data collection and analysis, decision to publish, or preparation of the manuscript.

**Competing interests:** The authors have declared that no competing interests exist.

**Abbreviations:** ACE2, angiotensin-converting enzyme 2; ARDS, acute respiratory distress syndrome; AT2, alveolar type II; BSL3, Biosafety Level 3; COVID-19, Coronavirus Disease 2019; DPI, days postinfection; hACE2, human ACE2; HE, hematoxylin–eosin; IACUC, Institutional Animal Care and Use Committee; ISH, in situ hybridization; MMZ, methimazole; OB, olfactory bulb; OE, olfactory epithelium; OSN, olfactory sensory neuron; PDPN, Podoplanin; PFU, plaque-forming unit; RE, respiratory epithelium; RFP, Red Fluorescent Protein; SARS-CoV-2, Severe Acute Respiratory Syndrome Coronavirus 2; SV40, simian virus 40; vWF, von Willebrand's Factor; WPRE, woodchuck hepatitis virus posttranscriptional regulatory element.

## Introduction

Understanding the cellular mechanisms that underlie Coronavirus Disease 2019 (COVID-19) is necessary to determine how to best prevent infection and treat affected individuals [1]. Severe Acute Respiratory Syndrome Coronavirus 2 (SARS-CoV-2) utilizes the transmembrane carboxypeptidase angiotensin-converting enzyme 2 (ACE2) as a host cell-surface receptor for viral entry. This interaction between virus and ACE2 has been defined by structural biochemical studies, cell culture, and transgenic mouse studies wherein the expression of human ACE2 (hACE2) is sufficient to confer infectability of parental SARS-CoV-2 strains [2–4].

ACE2 is a type 1 transmembrane protein that exists in vivo in two states: a cell-surface protein and a circulating cleaved "soluble" form primarily generated by the ADAM17 (aka TACE) sheddase [5,6]. While a central role for membrane-bound ACE2 is accepted, recent studies have suggested that soluble ACE2 can also mediate viral entry and infection, perhaps explaining the widespread tropism of SARS-CoV-2 in cell types that do not express detectable ACE2 such as endothelium, myocardium, and immune cells [7,8]. Clarifying and testing the in vivo significance of these mechanisms is translationally relevant given proposed COVID-19 therapies utilizing recombinant soluble ACE2 or blocking antibodies to competitively inhibit viral spread [9–13].

SARS-CoV-2 is a respiratory virus, and initial infection of epithelium in the nasal cavity and airways is thought to be ACE2-mediated. However, the role of ACE2 in mediating viral spread after initial infection remains untested. Multiple ACE2-independent mechanisms have been proposed including alternative receptors (e.g., DC-SIGN, L-SIGN, CD147, NRP1) as well as receptor-independent phenomena [14–18]. Testing the absolute requirement of ACE2 as infection progresses is critical to understanding of COVID-19 pathogenesis and would inform the design of targeted therapies.

The application of mouse models to investigate COVID-19 pathogenesis has been hindered by the fact that the wild-type SARS-CoV-2 spike protein is unable to bind the mouse ACE2 protein, a necessary first step in viral cellular entry and infection [1,2]. Several hACE2-expressing mouse models have been generated to investigate COVID-19 pathogenesis [19–22], but only the K18-hACE2 line confers severe illness like that observed in patients [4]. K18-hACE2 random transgenic mice express hACE2 in a nonendogenous fashion primarily restricted to epithelial cells and do not enable genetic dissection of viral pathogenesis. To identify cell-specific roles of ACE2 during SARS-CoV-2 infection, we generated new lines of mice in which hACE2 is expressed from the mouse *Ace2* locus in a manner that enables cell-specific loss of function (hACE2-floxed, hACE2$^{fl}$) and *Rosa26* mice in which hACE2 expression and gain of function can be conferred in a cell-specific manner (loxP-stop-loxP-hACE2, LSL-hACE2). Our results demonstrate that acute lung injury can occur in the absence of pulmonary infection and identify the olfactory epithelium (OE) and cerebral neurons as critical cellular sites of infection during lethal COVID-19. In all examined sites of infection—over multiple organs and diverse cell types—our genetic experiments illustrate an absolute, cell-autonomous requirement for ACE2 in viral entry.

## Results

### Generation and characterization of conditional hACE2 knock-in animals

To genetically investigate the cellular pathogenesis of COVID-19 in vivo, we generated gene targeted mice in which the coding sequence of the first translated exon of the mouse *Ace2* gene (exon 2) was replaced by an expression cassette including the human *ACE2* cDNA (Fig 1A). In the original generation of K18-hACE2 transgenic mice, it was noted that hACE2 expression

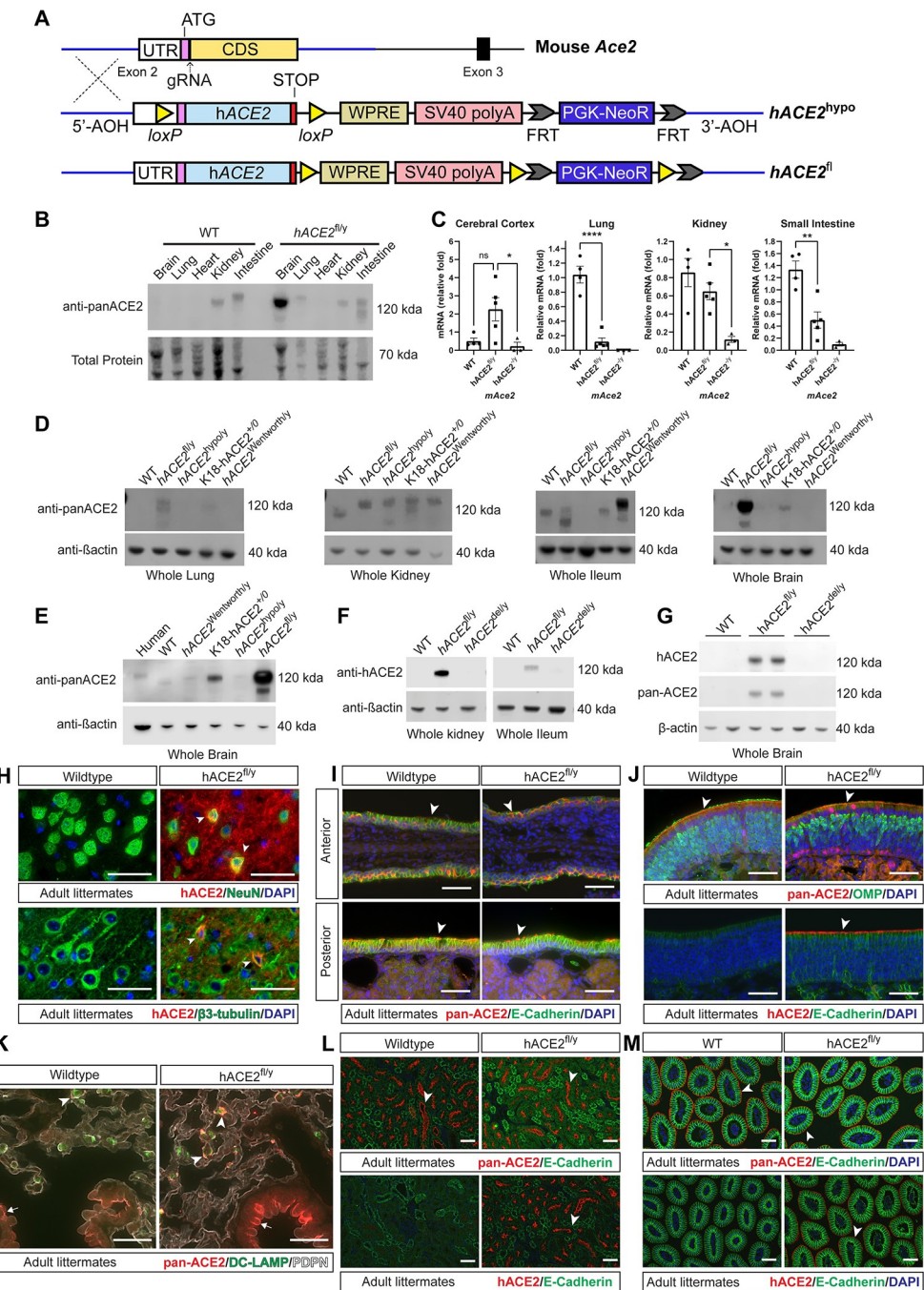

**Fig 1. Characterization of $hACE2^{fl/y}$ and $hACE2^{hypo}$ knock-in mice that express hACE2 in a conditional manner.**
(**A**) Generation of the $hACE2^{fl/y}$ and $hACE2^{hypo}$ alleles using gene targeting of the mouse *Ace2* locus. (**B**) Immunoblotting of whole tissue lysates from WT and $hACE2^{fl/y}$ organs using an antibody recognizing both mouse and human ACE2 (anti-panACE2). Total protein stain used to assess protein input. Representative of 3 independent experiments, *n* = 3 animals per genotype. (**C**) qPCR was performed using primers that detect untranslated 5′ mRNA sequences present in WT *Ace2*, $hACE2^{fl/y}$, and $hACE2^{hypo/y}$ alleles using the indicated tissues. (**D**) Immunoblotting of whole tissue lysates from WT, $hACE2^{fl/y}$, $hACE2^{hypo/y}$, K18-hACE2 transgenic, and hACE2 knock-in mice generated by the Wentworth lab ($hACE2^{Wentworth/y}$) was performed using anti-panACE2 and anti-β-actin (as a loading control). Representative of 3 independent experiments, *n* = 3 animals per genotype. (**E**) Immunoblotting of whole brain lysates from humans (Human) and the indicated mouse strains defined in (D) was performed using anti-panACE2 and b-actin antibodies. Representative of 3 independent experiments, *n* = 3 animals per genotype. (**F, G**) Immunoblotting of whole tissue lysates from WT, $hACE2^{fl/y}$, and $hACE2^{del/y}$ mice was performed using antibodies that specifically recognize hACE2 protein or anti-panACE2 antibodies. B-actin immunoblotting is shown as a loading control. Each

blot is representative of 4 independent experiments, *n* = 4 animals per genotype. (**H**) Immunohistochemistry of ACE2 using hACE2 antibodies and neurons using NeuN or β3-tubulin antibodies in WT and *hACE2*$^{fl/y}$ mouse cerebral cortex. Arrows indicate NeuN or β3-tubulin-positive neurons colocalized with ACE2 staining. Representative of *n* = 3 per genotype. Scale bars 50 μm. (**I**) Immunohistochemistry using pan-ACE2 and E-Cadherin antibodies is shown for WT and *hACE2*$^{fl/y}$ epithelium from the anterior and posterior part of the nasal cavity. Arrowheads indicate ACE2 + epithelial cells. *n* = 3 animals per genotype. (**J**) Immunohistochemistry using pan-ACE2 and hACE2 antibodies is shown for WT and *hACE2*$^{fl/y}$ OE costained with OMP (mature olfactory sensory neurons) or epithelial cadherin (E-cadherin, sustentacular cells). Arrowheads indicate ACE2+ olfactory epithelial cells. *n* = 3 animals per genotype. Scale bars 50 μm. (**K**) Immunostaining using pan-ACE2 antibodies is shown for WT and *hACE2*$^{fl/y}$ lung costained with antibodies recognizing DC-LAMP. Arrowheads indicate ACE2+ AT2 cells. Arrows indicate ACE2+ bronchial epithelial cells. *n* = 3 animals per genotype. Scale bars 50 μm. (**L**) Immunostaining using pan-ACE2 and hACE2 antibodies is shown for WT and *hACE2*$^{fl/y}$ kidney costained with E-cadherin (epithelium). Arrowheads indicate ACE2 + tubular epithelial cells. *n* = 3 animals per genotype. Scale bars 50 μm. (**M**) Immunostaining using pan-ACE2 and hACE2 antibodies is shown for WT and *hACE2*$^{fl/y}$ small intestine costained with E-cadherin (enterocytes). Arrowheads indicate ACE2+ enterocytes. *N* = 3 animals per genotype. Scale bars 50 μm. ns *p* > 0.05; *$^*$p* < 0.05; *$^{**}$p* < 0.01; *$^{****}$p* < 0.0001 by unpaired, two-tailed *t* test. Numerical data in corresponding S1 Metadata tab. hACE2, human ACE2; OE, olfactory epithelium; OMP, olfactory marker protein; WT, wild-type.

levels positively correlated with infectious disease severity [23]. Accordingly, since prior hACE2 knock-in alleles failed to confer lethal SARS CoV-2 infection [21,24], we inserted a woodchuck hepatitis virus posttranscriptional regulatory element (WPRE) and simian virus 40 (SV40) late polyA sequence to augment expression from the *Ace2* locus (Fig 1A). To enable conditional loss of hACE2 expression, we engineered one line of animals with loxP sites that flanked the hACE2 cDNA and SV40 poly cassette (hereafter termed "*hACE2*$^{hypo}$") (Fig 1A). Since the start codon in Exon 2 is preceded by interspecies conserved 5′ untranslated sequence that would be interrupted by the presence of a loxP sequence, we also generated a second knock-in line with loxP sites flanking the WPRE-polyA sequence to enable Cre-mediated loss of ACE2 expression without altering 5′ regulation of the *Ace2* locus (hereafter termed the "*hACE2*$^{fl}$" allele, with genotypes of *hACE2*$^{fl/y}$ in male animals or *hACE2*$^{fl/+}$ or *hACE2*$^{fl/fl}$ in female animals because the *Ace2* locus is on the X chromosome) (Fig 1A). To compare hACE2 protein expression in *hACE2*$^{hypo}$ and *hACE2*$^{fl}$ mice with endogenous mouse ACE2 (mACE2) protein expression alongside prior hACE2 mouse models, we performed western blotting of whole tissue lysates and immunostaining of tissue sections using pan-ACE2 antibodies that recognize both hACE2 and mACE2 proteins or hACE2-specific antibodies. Male mice were used because the *Ace2* allele is located on the X chromosome, enabling a straightforward comparison from the single expressed allele in all cell types. Western blotting with pan-ACE2 antibodies revealed that *hACE2*$^{hypo}$ mice expressed wild-type levels of hACE2 in the kidney but demonstrated no detectable expression in the brain, lung, or ileum. In contrast, *hACE2*$^{fl}$ mice expressed wild-type levels of ACE2 in heart, intestine, and kidney and higher than wild-type levels in the brain and lung (Fig 1B). Tissue-specific differences in hACE2 protein expression detected in *hACE2*$^{hypo}$ and *hACE2*$^{fl}$ mice corresponded to differences in hACE2-encoding mRNA levels assessed using qPCR (Figs 1C and S1A). We directly compared hACE2 protein levels in these new lines with two previously reported hACE2 mouse models: a K18-hACE2 transgenic line [4,23] and a hACE2 knock-in line developed by the Wentworth lab [25]. Western blotting was notable for elevated levels of hACE2 brain expression in K18-hACE2 and *hACE2*$^{fl}$ mice compared with those in the brains of wild-type mice, humans, Wentworth, or *hACE2*$^{hypo}$ knock-in lines (Fig 1D and 1E). To test the ability of Cre-mediated loss of the floxed WPRE-polyA cassette to block hACE2 protein expression, we crossed *hACE2*$^{fl}$ mice with *Sox2*-Cre animals to generate germline, Cre-recombined animals (hACE2$^{del/y}$). Western blotting with pan-ACE2 and hACE2 antibodies demonstrated complete loss of detectable hACE2 protein in all tissues (Fig 1F and 1G). The apparent molecular weight for hACE2 in *hACE2*$^{fl}$ mouse tissues varied, but all bands detected by anti-ACE2 antibodies were lost

following Cre-mediated recombination (Fig 1F and 1G), consistent with the presence of tissue-specific splice variants and/or posttranslational modification of the hACE2 protein.

To further compare knock-in hACE2 expression and localization to endogenous mACE2 expression, we performed immunostaining of tissue sections from *hACE2*fl, hACE2del/y, and wild-type mice with pan-ACE2 antibodies and antibodies specific for hACE2. hACE2 expression was sparsely detected in NeuN-positive and β3-tubulin-positive neurons in the brains of *hACE2*fl mice (Fig 1H) consistent with previous descriptions of brain ACE2 expression in mice and humans [26,27]. ACE2 was also detected in a discontinuous pattern on the apical brush border of E-cadherin expressing ciliated epithelial cells in the anterior and posterior nasal respiratory epithelium (RE) of both *hACE2*fl and wild-type mice (Fig 1I), consistent with prior reports of its localization in ciliated epithelial cells [28,29]. Immunostaining of the OE using pan-ACE2 and human-specific ACE2 antibodies also showed similar expression levels and an apical cellular expression pattern in control and *hACE2*fl animals (Fig 1J). ACE2 was sparsely detected at low levels in DC-LAMP-expressing alveolar type II (AT2) cells in the wild-type mouse lung, and at higher levels in AT2 cells in the lung of *hACE2*fl mice (Fig 1K) consistent with previous descriptions in mouse and human [30–32]. Staining of the kidney and intestine revealed specific epithelial cell expression in *hACE2*fl animals identical to that in wild-type mice (Fig 1L and 1M). Lastly, immunostaining of *hACE2*del/y tissues demonstrated loss of apical epithelial hACE2 expression in the intestine and kidney compared to littermate *hACE2*fl/y tissues (S1B Fig).

These studies demonstrate that the *hACE2*fl allele, but not the *hACE2*hypo allele, expresses hACE2 across multiple tissues in a pattern like that of the endogenous mouse *Ace2* allele, albeit with higher expression in the lung and brain. They also demonstrate loss of detectable hACE2 protein expression from the *hACE2*fl allele following Cre-mediated removal of the WPRE-polyA cassette.

## hACE2fl mice experience lethal infection following exposure to SARS-CoV-2

A subset of human patients exposed to SARS-CoV-2 experience severe illness that leads to rapid respiratory failure and death [33,34]. This lethal disease course has been reproduced in K18-hACE2 transgenic mice [4], but not in other animal models such as hamsters or previously reported hACE2 knock-in mice [19,21,25,35,36]. To define the disease response of *hACE2*fl and *hACE2*hypo mice, we exposed *hACE2*fl/y, hACE2del/y, and *hACE2*hypo mice to $10^5$ PFU of SARS-CoV-2 virus via nasal inhalation (Fig 2A). *hACE2*fl/y animals exhibited weight loss beginning 2 days after viral exposure, appeared hunched and sluggish by day 4 with tachypnea and increased work of breathing, then died or were killed by 6 days postinfection due to >20% weight loss (humane endpoint). In contrast, infected *hACE2*del/y mice and *hACE2*hypo mice exhibited no weight loss or respiratory distress at time points up to 14 days postinfection (Fig 2B–2E). Similar findings were obtained following exposure to $>10^2$ PFU of SARS-CoV-2 virus, and following infection of *hACE2*fl/y animals in two distinct ABSL3 facilities (S2 Fig). Like *hACE2*hypo mice described here, the previously reported Wentworth line of hACE2 conditional knock-in animals [25] expressing lower levels of hACE2 than *hACE2*fl/y mice (Fig 1D) failed to exhibit significant weight loss or lethality after exposure to SARS-CoV-2 (Figs 2G and 3F). These studies demonstrate that *hACE2*fl/y animals develop reproducible, lethal disease after exposure to SARS-CoV-2, while other hACE2 knock-in animals with lower levels of hACE2 expression in the brain and lung do not.

Human studies have revealed a sex difference in COVID-19 susceptibility, with male sex identified as an independent risk factor for severe illness and death [37,38] and, more recently, a reduced susceptibility to lethal COVID-19 following infection with the Omicron BA.1

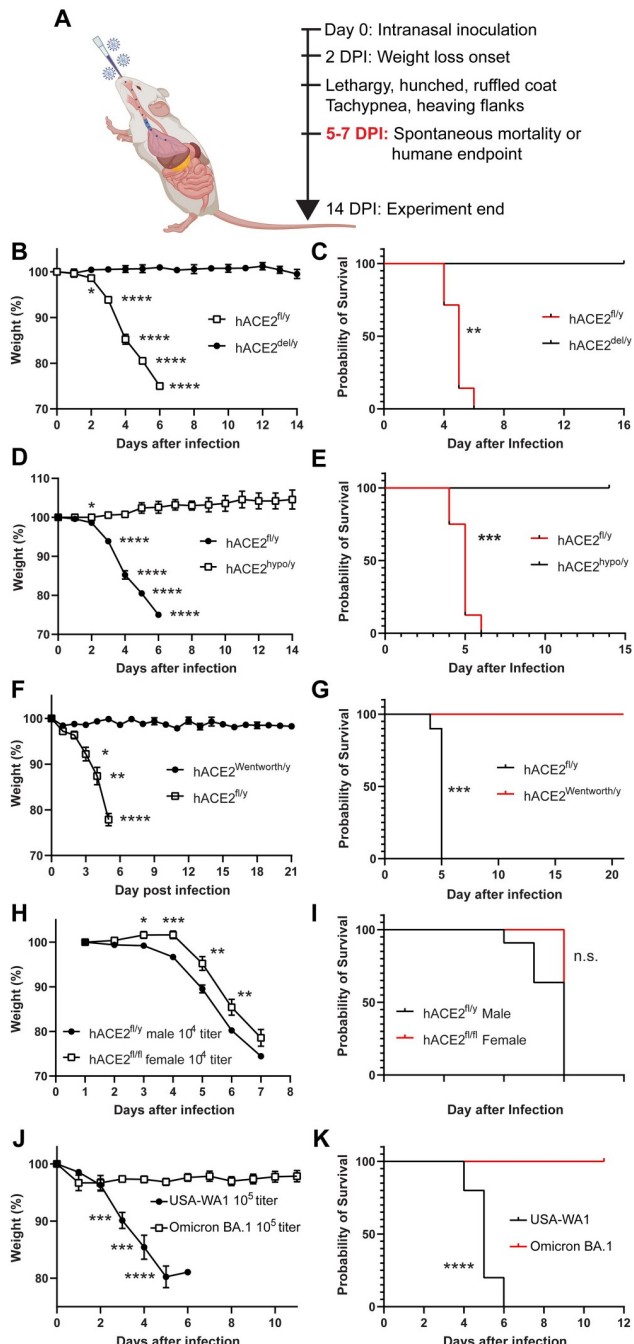

**Fig 2. Intranasal SARS-CoV-2 infection of *hACE2*$^{fl/y}$ mice confers weight loss and lethality reversed by Cre recombination.** (**A**) Experimental design for acute infection with SARS-CoV-2 virus. (**B, C**) Weight loss and survival of *hACE2*$^{fl/y}$ and *hACE2*$^{del/y}$ mice after infection with $10^5$ viral titer per mouse. $n = 8$ for both genotypes, two independent experiments. (**D, E**) Weight loss and survival of *hACE2*$^{fl/y}$ and *hACE2*$^{hypo/y}$ mice after infection with $10^5$ viral titer per mouse. $n = 8$ for both genotypes, two independent experiments. (**F, G**) Weight loss and survival of *hACE2*$^{fl/y}$ and *hACE2*$^{Wentworth/y}$ mice after infection with $10^5$ PFU of SARS-CoV-2 virus. $n = 8$ (*hACE2*$^{fl/y}$) and 5 (*hACE2*$^{Wentworth/y}$), one experiment. (**H, I**) Weight loss and survival of littermate male and female *hACE2*$^{fl/y}$ mice after infection with $10^4$ titer of SARS-CoV-2 virus. $n = 11$ (male) and 6 (female), one experiment. (**J, K**) Weight loss and survival of littermate male *hACE2*$^{fl/y}$ mice after infection with $10^5$ titer of the USA-WA1 or Omicron BA.1 strains of SARS-CoV-2 virus. $n = 10$ (USA-WA1) and 7 (Omicron BA.1), two experiments. **Note:** Data for *hACE2*$^{fl/y}$ infections at $10^5$ titer are reused throughout panels (B-E) in this figure as littermate controls were not applicable and infections done contemporaneously with the same viral stock. ns $p > 0.05$; $^*p < 0.5$; $^{**}p < 0.01$; $^{***}p < 0.001$; $^{****}p < 0.0001$ by multiple unpaired two-tailed *t* test or log-rank Mantel Cox test. Numerical data in corresponding S1 Metadata tab. DPI, days postinfection; hACE2, human ACE2; SARS-CoV-2, Severe Acute Respiratory Syndrome Coronavirus 2.

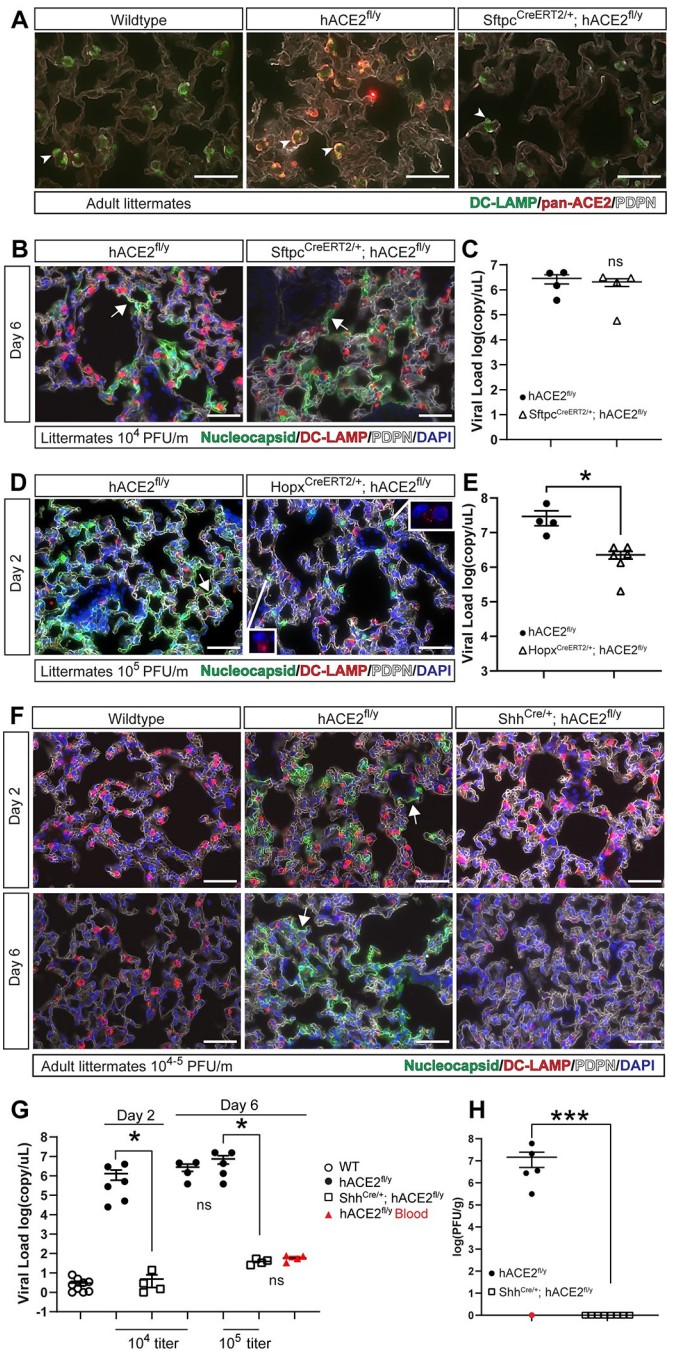

**Fig 3. SARS-CoV-2 infects lung alveolar type I cells in an ACE2-dependent manner.** (**A**) Immunodetection of ACE2 using pan-ACE2 antibodies and DC-LAMP in wild-type $hACE2^{fl/y}$, and Sfptc$^{CreERT2/+}$; $hACE2^{fl/y}$ mouse lungs 6 days after infection with $10^4$ PFU of SARS-CoV-2. Arrowheads indicate DC-LAMP+ AT2 cells. $n = 4$ for both genotypes, one experiment. (**B**) Immunohistochemistry of $hACE2^{fl/y}$ and Sfptc$^{CreERT2/+}$; $hACE2^{fl/y}$ mouse lungs 6 days after infection with $10^4$ PFU of SARS-CoV-2 virus was performed using antibodies that recognize SARS-CoV-2 nucleocapsid, DC-LAMP (AT2 cells), and PDPN (AT1 cells) as well as the nuclear stain DAPI. Arrows indicate nucleocapsid colocalized with PDPN. $N = 4$ for both genotypes, one experiment. (**C**) qPCR was performed on $hACE2^{fl/y}$ and Sfptc$^{CreERT2/+}$; $hACE2^{fl/y}$ mouse lungs harvested 6 days after infection with $10^4$ PFU of SARS-CoV-2 to measure total viral load. $N = 4$ for both genotypes, one experiment. (**D**) Immunohistochemistry of $hACE2^{fl/y}$ and Hopx$^{CreERT2/+}$; $hACE2^{fl/y}$ mouse lungs 2 days after infection with $10^5$ PFU of SARS-CoV-2 virus was performed using antibodies that recognize SARS-CoV-2 nucleocapsid, DC-LAMP, and PDPN as well as the nuclear stain DAPI. Arrows indicate nucleocapsid colocalized with PDPN in AT1 cells. Arrowheads indicate nucleocapsid colocalized with DC-LAMP in AT2 cells. The boxed regions show DC-LAMP staining (red) only in the indicated nucleocapsid positive (green) AT2

cells because the dim DC-LAMP signal is obscured by nucleocapsid signal when both are visible. $N = 4$–5 for both genotypes, one experiment. (**E**) qPCR was performed on $hACE2^{fl/y}$ and $Hopx^{CreERT2/+}$; $hACE2^{fl/y}$ mouse lungs harvested 2 days after infection with $10^5$ PFU of SARS-CoV-2 to measure total viral load. $N = 4$–6 for both genotypes, one experiment. (**F**) Immunohistochemistry of $hACE2^{fl/y}$ and $Shh^{Cre/+}$; $hACE2^{fl/y}$ mouse lungs 2 and 6 days after infection with $10^4$ or $10^5$ PFU of SARS-CoV-2 virus was performed using antibodies that recognize SARS-CoV-2 nucleocapsid, DC-LAMP (AT2 cells), and PDPN (AT1 cells) as well as the nuclear stain DAPI. $N = 6$ for all genotypes, two independent experiments. Arrows indicate nucleocapsid colocalized with PDPN. (**G**) qPCR was performed on $hACE2^{fl/y}$ and $Shh^{Cre/+}$; $hACE2^{fl/y}$ mouse lungs harvested 2 and 6 days after infection with $10^4$ or $10^5$ PFU of SARS-CoV-2 to measure total viral RNA load. Simultaneous measurement using whole blood was performed 6 days after infection (shown in red) to measure circulating levels. (**H**) Infectious viral load was measured by PFU from $hACE2^{fl/y}$ and $Shh^{Cre/+}$; $hACE2^{fl/y}$ mouse lungs 6 days after infection with $10^5$ PFU of SARS-CoV-2. $N > 6$ for both genotypes. **Note:** $hACE2^{fl/y}$ $10^4$ titer data for 3C and 3G are the same. Scale bars in all images 50 μm. $^{***}p < 0.001$; $^{*}p < 0.05$; ns $p > 0.05$, significance determined by unpaired two-tailed $t$ test. Numerical data in corresponding S1 Metadata tab. ACE2, angiotensin-converting enzyme 2; AT1, alveolar type 1; AT2, alveolar type 2; PDPN, Podoplanin; PFU, plaque forming assay; SARS-CoV-2, Severe Acute Respiratory Syndrome Coronavirus 2.

SARS-CoV-2 variant in comparison to earlier viral variants such as the USA-WA1 variant used in this study [39,40]. Consistent with these human findings, although severe weight loss and 100% mortality were observed in both groups following exposure to the more virulent USA-WA1 virus, male animals exhibited symptoms at earlier time points than females (Fig 2H and 2I). Also consistent with prior human and rodent studies [41], infection of $hACE2^{fl/y}$ animals with $10^5$ PFU of the Omicron BA.1 variant failed to confer lethal disease (Fig 2J and 2K). Thus, although severe, lethal disease is observed in virtually all $hACE2^{fl/y}$ animals compared with a rare fraction of human patients, $hACE2^{fl/y}$ animals exhibit sex- and strain-specific responses to SARS-CoV-2 infection that reproduce those observed in the human population.

## Epithelial cell hACE2 expression is required for pulmonary SARS-CoV-2 infection

Following infection with SARS-CoV-2, $hACE2^{fl/y}$ animals exhibited labored breathing, indicative of respiratory distress and COVID-19 lung disease. Prior studies using single-cell RNAseq and immunohistochemical analysis of mouse and human lungs and our immunostaining of $hACE2^{fl}$ mouse lungs identified AT2 cells as ACE2-positive, suggesting that AT2 cell infection might play a key role in COVID-19 pneumonia [30–32,42]. Therefore, we used the $Sftpc^{CreERT2}$ knock-in allele to test whether ACE2-dependent infection of AT2 cells is required for lung infection in $hACE2^{fl}$ animals. Following tamoxifen induction to activate Cre recombination, $Sftpc^{CreERT2}$; $hACE2^{fl/y}$ mice exhibited loss of hACE2 protein in lung AT2 cells (Fig 3A). However, tamoxifen-treated $Sftpc^{CreERT2}$; $hACE2^{fl/y}$ males infected with SARS-CoV-2 virus exhibited levels of viral nucleocapsid protein and viral RNA in the lung indistinguishable from those in $hACE2^{fl/y}$ animals (Fig 3B and 3C). These findings suggested that cell types other than AT2 cells can support SARS-CoV-2 infection of the lung.

Histologic examination of $hACE2^{fl/y}$ lungs 2 and 6 days after infection with SARS-CoV-2 revealed viral nucleocapsid staining primarily along the cell membrane of Podoplanin (PDPN)-positive alveolar type 1 (AT1) epithelial cells (Fig 3B, 3D, and 3F). To specifically test the role of ACE2 in AT1 cells, we next examined SARS-CoV-2 infection of $Hopx^{CreERT2}$; $hACE2^{fl/y}$ animals in which Cre is active in AT1 but not AT2 cells when Cre activity is activated by tamoxifen administration in mature mice [43,44]. SARS-CoV-2 nucleocapsid was detected in DC-LAMP-expressing AT2 cells but not in PDPN-expressing AT1 cells in the alveoli of infected $Hopx^{Cre}$; $hACE2^{fl/y}$ animals (Fig 3D). $Hopx^{CreERT2}$; $hACE2^{fl/y}$ animals infected with SARS-CoV-2 displayed significant but incomplete reductions in viral nucleocapsid protein and viral RNA in the lung (Fig 3D and 3E), consistent with infection of AT2 (Fig 3D) and bronchial epithelial cells (S3 Fig). Finally, to test the role of ACE2 in AT1, AT2, and

bronchiolar epithelial cell infection in the lung, we examined SARS-CoV-2 infection of Shh[Cre]; hACE2[fl/y] animals in which Cre is active in all lower respiratory and gastrointestinal epithelium ([45]; S4 and S5 Figs). SARS-CoV-2 infection was blocked completely in Shh[Cre]; hACE2[fl/y] lungs, as assessed by immunostaining for viral nucleocapsid (Fig 3F), measurement of lung viral load using both qPCR for viral RNA (Fig 3G) and culture of viral plaque-forming units (PFUs) from freshly harvested lung tissue (Fig 3H). On day 6 postinfection, a low level of virus was detected using qPCR that was equivalent to the level detected in circulating blood (Fig 3G). These findings demonstrate that lung infection by SARS-CoV-2 occurs in both AT1 and AT2 epithelial cells in a cell-autonomous, ACE2-dependent manner.

## Lung epithelial infection is not required for lethal COVID-19 in hACE2[fl/y] animals

Studies of human patients with severe COVID-19 pneumonia have revealed evidence of both lung infection by SARS-CoV-2 virus [32,42,46,47] and acute respiratory distress syndrome (ARDS), manifest by noncardiogenic pulmonary edema, severe hypoxemia, and a systemic inflammatory response [48,49]. The extent to which these two mechanisms of lung disease are interconnected, and their respective roles in COVID-19 mortality, have been difficult to define clinically in humans. Following exposure to $10^4$ or $10^5$ PFU per mouse of SARS-CoV-2, both hACE2[fl/y] and Shh[Cre]; hACE2[fl/y] animals exhibited weight loss and lethality (Figs 4A, 4B, S6A, and S6B). There was a slight delay in the onset of symptoms in Shh[Cre]; hACE2[fl/y] animals, but both hACE2[fl/y] and Shh[Cre]; hACE2[fl/y] animals were severely hypoxemic 5–6 days after infection, with oxygen ($O_2$) saturations (SpO2) ranging between 70% and 85%, levels like those of severely ill patients requiring mechanical ventilation (Fig 4C). Hematoxylin–eosin (HE) staining of lung sections from wild-type, hACE2[fl/y], and Shh[Cre]; hACE2[fl/y] animals 6 days after exposure to SARS-CoV-2 revealed alveolar consolidation, interstitial thickening, and the presence of focal infiltrates and hemorrhage in both hACE2[fl/y] and Shh[Cre]; hACE2[fl/y] animals that were not observed in wild-type controls (Fig 4D). Virtually identical results were obtained following exposure to $10^5$ PFU of SARS-CoV-2 (S6 Fig). SARS-CoV-2-infected hACE2[fl/y] and Shh[Cre]; hACE2[fl/y] animals also exhibited thrombus-filled pulmonary vessels that were not observed in the lungs of wild-type animals exposed to virus (Figs 4D, 4E, and S6D), a prominent finding also observed in human lungs harvested from individuals with lethal COVID-19 disease [48].

Consistent with a global lung inflammatory state, we observed uniformly elevated expression of the inflammation-induced proteins ICAM1 and PDPN (aka RTI40) in the alveolar epithelial cells of infected hACE2[fl/y] and Shh[Cre]; hACE2[fl/y] animals compared with SARS-CoV-2 inoculated wild-type controls (Fig 4E) [50]. Expression of the pro-coagulant, inflammation-induced protein von Willebrand's Factor (vWF) was also up-regulated in the lung capillary endothelial cells of SARS-CoV-2-infected hACE2[fl/y] and Shh[Cre]; hACE2[fl/y] animals compared with wild-type controls (Fig 4F), consistent with the presence of intravascular thrombi in those lungs. These studies and those shown in Fig 3 demonstrate that acute lung injury and hypoxemic respiratory failure may arise in the absence of primary SARS-CoV-2 lung infection and implicate an inflammatory state and ARDS due to extrapulmonary infection as an important mechanism of respiratory failure and lethality in hACE2[fl/y] mice.

## Lethal COVID-19 in hACE2[fl/y] mice is associated with SARS-CoV-2 infection of the olfactory epithelium, olfactory bulb, and cerebrum

The above results suggested that SARS-CoV-2 infection at cellular sites outside the lung that lack Shh[Cre] activity were sufficient to confer severe disease in hACE2[fl/y] animals. To define the

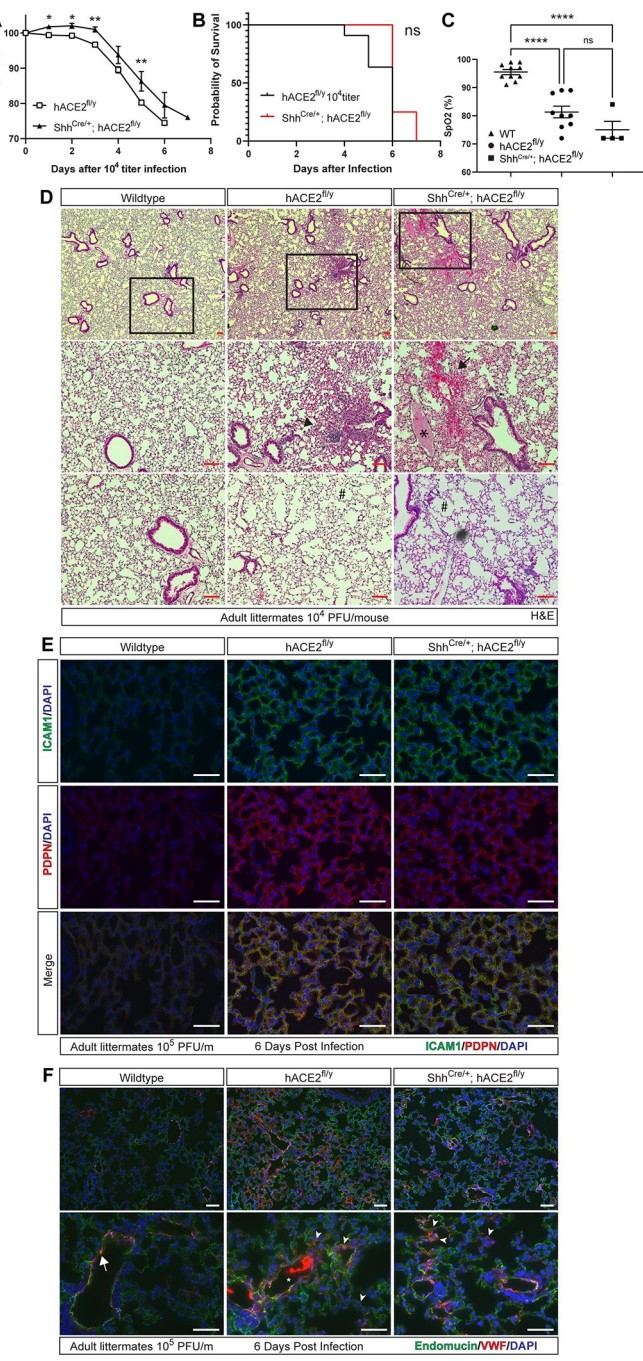

**Fig 4. hACE2 mice develop acute lung injury and hypoxemia in the absence of SARS-CoV-2 lung infection. (A, B)**
Weight loss and survival of $hACE2^{fl/y}$ and $Shh^{Cre/+}$; $hACE2^{fl/y}$ mice after infection with $10^4$ PFU of SARS-CoV-2. $N =$ 11 ($hACE2^{fl/y}$) and 4 ($Shh^{Cre/+}$; $hACE2^{fl/y}$), two independent experiments. **Note**: Data for $hACE2^{fl/y}$ same as Figs 2H and S2 since $Shh^{Cre/+}$; $hACE2^{fl/y}$ animals were littermates. (**C**) Pulse oximetry measured in WT, $hACE2^{fl/y}$, and $Shh^{Cre/+}$; $hACE2^{fl/y}$ mice 6 days after exposure to $10^4$ PFU of SARS-CoV-2 virus. (**D**) HE staining of WT, $hACE2^{fl/y}$, and $Shh^{Cre/+}$; $hACE2^{fl/y}$ lung tissue 6 days after exposure to $10^4$ PFU of SARS-CoV-2 virus. Boxed regions at higher magnification in images below. Arrows, sites of focal consolidation. Asterisks, intravascular thrombi. Hashtag, acute emphysematous changes. Representative of $N = 4$ animals per genotype. (**E, F**) Immunohistochemistry of WT, $hACE2^{fl/y}$, and $Shh^{Cre/+}$; $hACE2^{fl/y}$ lung tissue 6 days after exposure to $10^4$ PFU of SARS-CoV-2 virus using antibodies against ICAM-1 and PDPN, or vWF and Endomucin (endothelial cells). Arrowheads in F identify vWF-positive microvasculature of the lung in $hACE2^{fl/y}$ and $Shh^{Cre/+}$; $hACE2^{fl/y}$ animals. Representative of $N = 4$ animals per genotype. Scale bars in all images, 50 μm. **Note:** Images in each panel were taken at lower or higher magnification from the same tissue section respective to genotype. $^*p < 0.05$, $^{**}p < 0.001$; $^{****}p < 0.0001$ by unpaired two-tailed $t$ test,

one-way ANOVA with Holm–Sidak correction for multiple comparisons, or log-rank Mantel Cox test. Numerical data in corresponding S1 Metadata tab. HE, hematoxylin–eosin; ICAM-1, intracellular adhesion marker 1; PDPN, Podoplanin; SARS-CoV-2, Severe Acute Respiratory Syndrome Coronavirus 2; vWF, von Willebrand's Factor; WT, wild-type.

extrapulmonary sites of infection in Shh$^{Cre}$; *hACE2*$^{fl/y}$ mice that might be responsible for pulmonary inflammation and respiratory failure, we performed lineage tracing of the Shh$^{Cre}$ transgene along the path of inhaled virus using the R26-LSL-Red Fluorescent Protein (LSL-RFP, aka Ai14) Cre reporter allele. Shh$^{Cre}$ activity was detected uniformly in epithelial cells lining the trachea, lung airways, and alveoli, as well as in the transition zone epithelium that lies between the nasal cavity RE and OE (S4 Fig). In contrast, the RE and OE of the nasal passages failed to display Shh$^{Cre}$ activity (S4B Fig), suggesting that these sites of high ACE2 expression may be responsible for the lethal response to SARS-CoV-2 infection in *hACE2*$^{fl/y}$ mice.

Therefore, we compared SARS-CoV-2 nucleocapsid staining in the RE, OE, and adjacent olfactory bulb (OB) of the brain during infection in wild-type, *hACE2*$^{fl/y}$, and Shh$^{Cre}$; *hACE2*$^{fl/y}$ animals. Two days after infection, abundant SARS-CoV-2 nucleocapsid was observed in the RE and OE of *hACE2*$^{fl/y}$ and Shh$^{Cre}$; *hACE2*$^{fl/y}$ animals, but none was detected in the adjacent OB and cerebral cortex of the brain (Fig 5A and 5C). In contrast, 5 to 6 days after infection, relatively little SARS-CoV-2 nucleocapsid was detected in the RE and OE of *hACE2*$^{fl/y}$ and Shh$^{Cre}$; *hACE2*$^{fl/y}$ animals, although abundant signal was detected in the neighboring OB, cerebral cortex, and hippocampus (Figs 5B, 5C, and S7). Costaining with the nuclear neuronal marker NeuN and the glial cell marker GFAP revealed SARS-CoV-2 nucleocapsid specifically in neurons and not associated glial cells and demonstrated the presence of a reactive gliosis 5 days postinfection (Fig 5D). Sparse neuronal cell SARS-CoV-2 nucleocapsid and a reactive gliosis was also detected in the brainstem, a site more distant from the OE, but not in the cerebellum, 5 days after infection (S7 Fig). Consistent with the immunostaining studies to detect virus described above, in situ hybridization of the nasal cavity and adjacent tissues using RNAscope probes for SARS-CoV-2 RNA revealed little or no virus in the OE 5 days after infection with abundant viral RNA detected in the adjacent OB and cerebrum of the brain (Fig 5E and 5F). Unlike the OE, SARS-CoV-2 nucleocapsid was not detected in the choroid plexus or adjacent neurons at an early time point, 2 days after infection (S8A Fig). However, patches of neuronal SARS-CoV-2 nucleocapsid staining were observed 5 days after viral infection in proximity to the choroid plexus and meninges in both *hACE2*$^{fl/y}$ and Shh$^{Cre}$; *hACE2*$^{fl/y}$ animals (S8B Fig). HE staining of the choroid plexus along the lateral ventricle and cerebral cortex vasculature revealed immune cell infiltrates at 5 (but not 2) days after SARS-CoV-2 infection (S8C and S8D Fig), indicative of an inflammatory response like that described in humans [51,52].These findings are consistent with prior studies demonstrating that olfactory epithelial cells may rapidly clear the virus after infection with SARS-CoV-2 in hamsters [53] and in humans [28] and demonstrate that early infection of the OE is followed by later infection of neurons in the brain of *hACE2*$^{fl/y}$ mice.

## Pharmacologic ablation of the OE or genetic loss of hACE2 in the OE and neurons prevents lethal COVID-19 in hACE2$^{fl/y}$ mice

The studies described above indicated that SARS-CoV-2 infection of the OE rather than the lung is associated with weight loss and lethality. Furthermore, it afforded the opportunity to stringently test the requirement for ACE2 during initial infection and subsequent viral spread in vivo. Since genetic tools to drive Cre expression selectively in the OE are not available, we

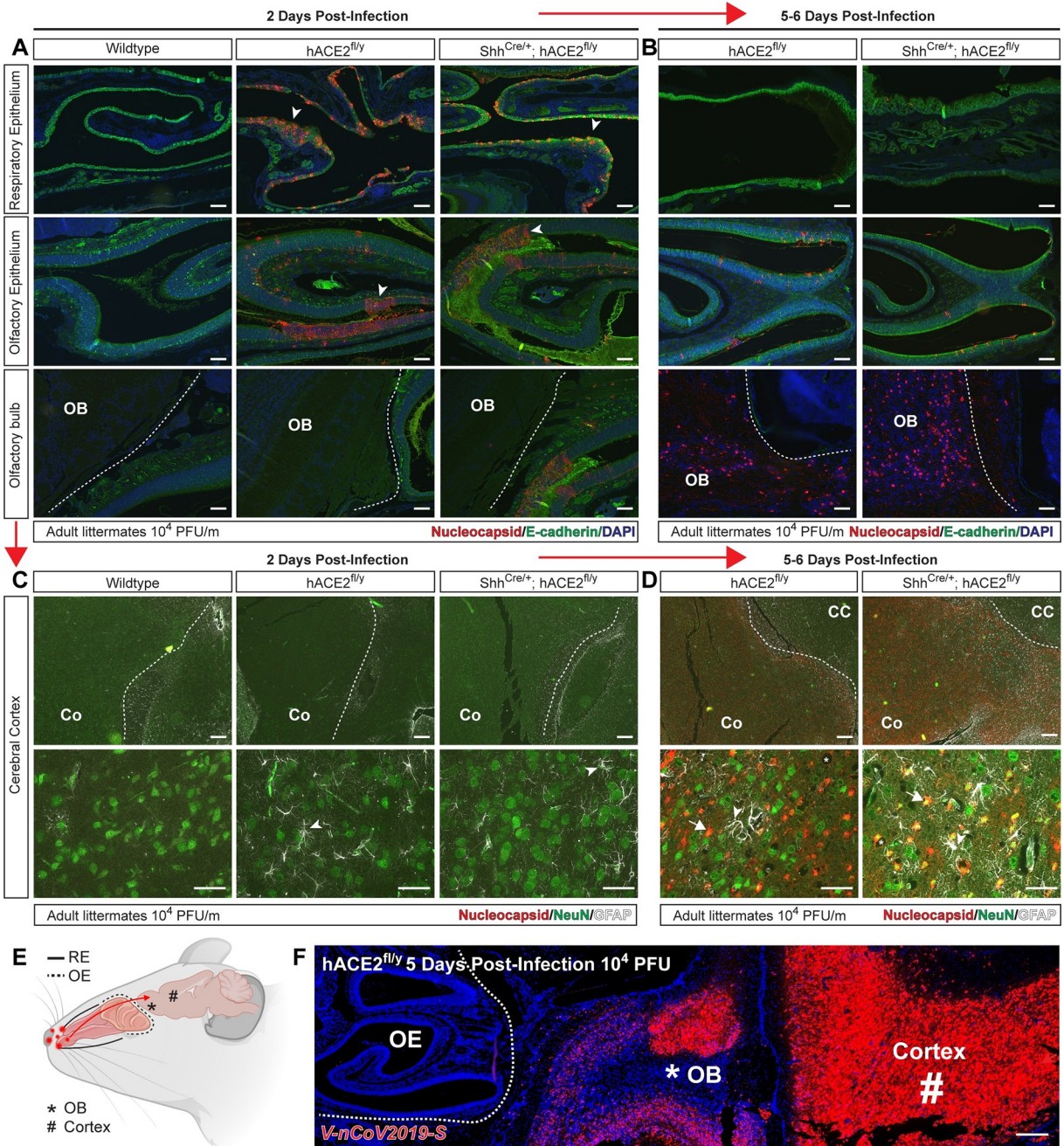

**Fig 5. SARS-CoV-2 infection of the OE and brain in hACE2^fl mice.** (**A, B**) Immunohistochemistry of SARS-CoV-2 nucleocapsid and epithelial cell E-cadherin in the RE, OE, and OB 2 and 5–6 days after infection of *hACE2*^fl/y and Shh^Cre/+; *hACE2*^fl/y mice. Arrowheads indicate sites of viral nucleocapsid detection. Representative of $N = 4$ animals per genotype and time point. Scale bars 100 μm. (**C, D**) Immunohistochemistry of SARS-CoV-2 nucleocapsid, neuronal NeuN, and glial cell GFAP in the cerebral cortex (Co) 2 and 5–6 days after infection. Arrowheads indicate sites of GFAP+ reactive gliosis. Arrows indicate nucleocapsid colocalization with NeuN staining. Representative of $N = 4$ animals per genotype and time point. Scale bars 100 μm top, 50 μm bottom. (**E**) Diagram of the mouse nasal cavity and cranial anatomy. (**F**) In situ hybridization detection of SARS-CoV-2 mRNA 5 days postinfection reveals virus in the OB and cerebral cortex of the brain, but not the OE of the nose. Scale bar 250 μm. **Note:** Images in each panel were taken at lower or higher magnification from the same tissue section respective to genotype and highlight different anatomical regions. OB, olfactory bulb of the brain; OE, olfactory epithelium; RE, respiratory epithelium; SARS-CoV-2, Severe Acute Respiratory Syndrome Coronavirus 2.

tested the requirement for OE infection using pharmacologic ablation of olfactory epithelial cells with methimazole (MMZ), a compound that is highly and specifically toxic for the OE in rodents [54]. HE staining of the OE 24 hours after treatment of $hACE2^{fl/y}$ mice with MMZ revealed loss of almost all olfactory epithelial cells (Fig 6A). In contrast, the underlying OB and lung did not show evidence of cell loss, cell damage, or inflammation (Fig 6A). We next tested whether ablation of the OE with MMZ impacts the clinical course of $hACE2^{fl/y}$ mice infected with SARS-CoV-2 virus. Remarkably, pretreatment with MMZ 24 hours prior to intranasal inoculation of SARS-CoV-2 prevented both weight loss and death in the majority of $hACE2^{fl/y}$ mice (Fig 6B and 6C). Immunostaining of the brain for SARS-CoV-2 nucleocapsid protein 6 days after infection revealed no detectable virus in the cerebral cortex of MMZ-treated $hACE2^{fl/y}$ mice compared with vehicle-treated $hACE2^{fl/y}$ mice (Fig 6D). In contrast, similar expression of SARS-CoV-2 nucleocapsid was detected in the lungs of vehicle-treated and MMZ-treated $hACE2^{fl/y}$ mice (Fig 6E), a finding consistent with the lack of lethality associated with isolated lung infection in $Shh^{Cre}$;LSL-hACE2$^{+/0}$ mice (Fig 8, described below). Analysis of MMZ-treated $hACE2^{fl/y}$ mice 14 days after SARS-CoV-2 infection revealed no viral nucleocapsid in either the brain or lung (Fig 6F and 6G), consistent with sparing of the brain and resolution of lung infection in surviving MMZ-treated $hACE2^{fl/y}$ mice. Consistent with brain infection being a driver of lethality, analysis of a single $hACE2^{fl/y}$ mouse that became ill and was euthanized 8 days after infection revealed strong staining for nucleocapsid in the brain and lung (S9 Fig).

To further test the requirement for SARS-CoV-2 infection of the OE and brain for lethal COVID-19 in $hACE2^{fl/y}$ mice, we generated Foxg1$^{Cre/+}$; $hACE2^{fl/y}$ mice in which Cre recombinase is expressed in the RE, OE, and neurons of the forebrain, but not in cells of the lung [55] (Figs 6H and S10), and K14-Cre; $hACE2^{fl/y}$ mice in which Cre recombinase is expressed in RE and transition zone epithelial cells of the nasal cavity (Keratin-14; S11A Fig). Following exposure to SARS-CoV-2, both $hACE2^{fl/y}$ and K14-Cre; $hACE2^{fl/y}$ mice exhibited weight loss and lethality associated with severe hypoxemia (S11B–S11D Fig), although K14-Cre; $hACE2^{fl/y}$ mice demonstrated absence of SARS-CoV-2 nucleocapsid staining in the nasal RE, consistent with a lack of infection at that site (S11E Fig). In contrast, Foxg1$^{Cre/+}$; $hACE2^{fl/y}$ mice survived and did not experience weight loss or hypoxemia (Fig 6I–6K). Consistent with these findings, Foxg1$^{Cre/+}$; $hACE2^{fl/y}$ mice exhibited evidence of SARS-CoV-2 infection in the lung but not the OE or RE at 2 days postinfection, and no evidence of brain infection at 6 days postinfection (S12 Fig). These pharmacologic and genetic findings are highly concordant and identify SARS-CoV-2 infection of the OE and brain as required for lethal COVID-19 in $hACE2^{fl/y}$ mice.

## Neuronal SARS-CoV-2 infection is required for lethal respiratory failure in hACE2$^{fl}$ mice

The genetic and pharmacologic studies described above suggested that SARS-CoV-2 infection of OE and/or neurons was required for lethal respiratory disease in hACE2$^{fl}$ animals. To further identify the infected cell type responsible for lethal disease, we tested whether hACE2 was required selectively in neurons for SARS-CoV-2 infection using the Baf53b (aka Actl6b)-Cre transgenic line that expresses Cre specifically in neurons but not in olfactory or other epithelial cells [56]. Importantly, lineage tracing studies demonstrated highly specific activity of the Baf53b-Cre allele in neurons, including olfactory sensory neurons (OSNs), but not in sustentacular cells or respiratory epithelial cells (S13 Fig). In contrast to $hACE2^{fl/y}$ littermates, following exposure to SARS-CoV-2, Baf53b-Cre; $hACE2^{fl/y}$ mice exhibited no signs of distress, did not lose weight or become hypoxemic, and had normal survival (Fig 7A–7C). Immunoblot

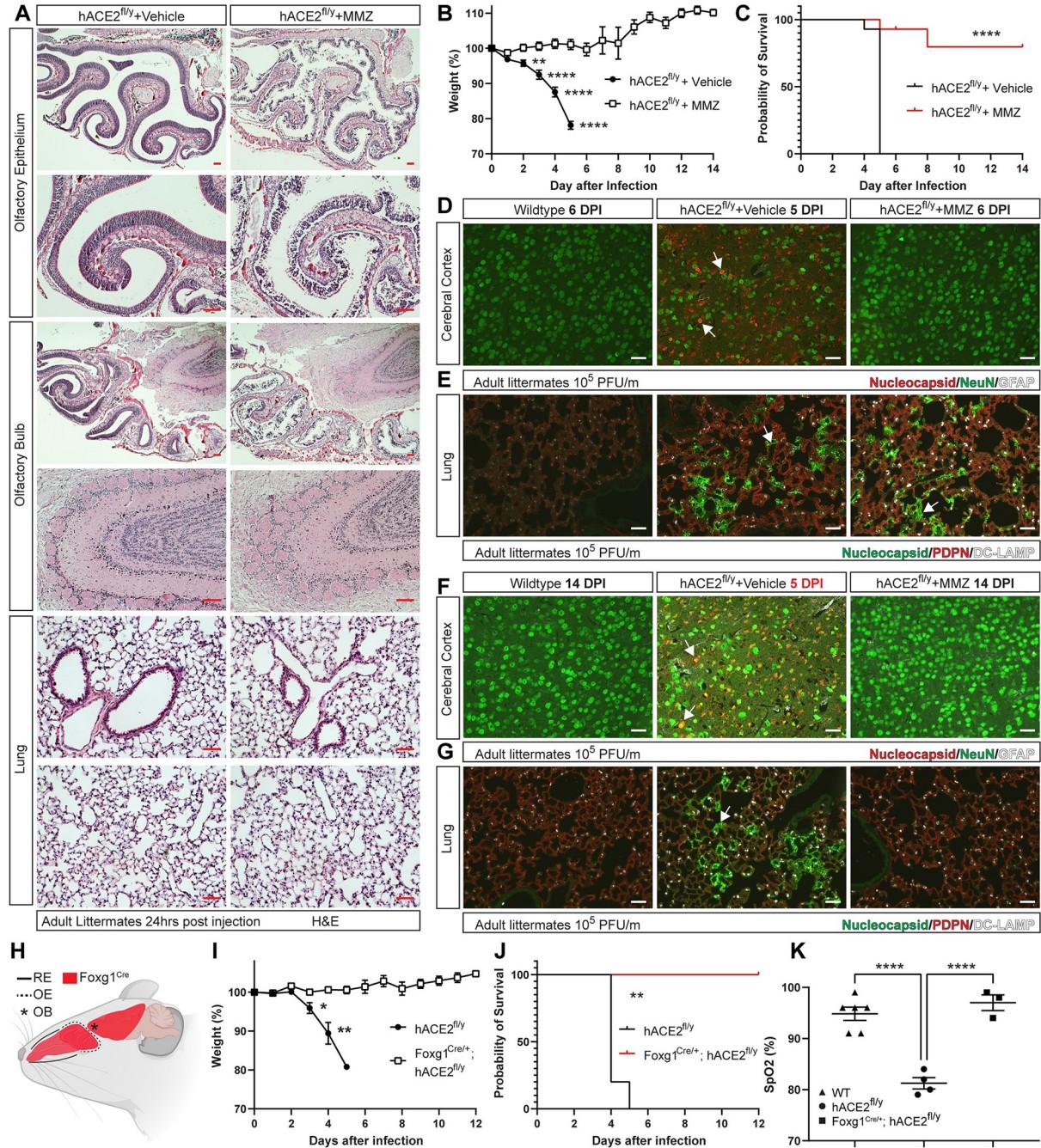

**Fig 6. MMZ ablation of the OE or genetic loss of hACE2 in OE and neurons prevents brain infection and lethal SARS-CoV-2 infection.**
(**A**) HE staining of the indicated mouse tissues was performed 24 hours after intraperitoneal injection of 100 mg/kg MMZ or vehicle control.
Scale bars 100 μm for the OE, OB, and 50 μm for the lung. Representative of $N = 3$ per condition. (**B, C**) Weight loss and survival of $hACE2^{fl/y}$
mice treated with MMZ or vehicle prior to infection with $10^5$ PFU of SARS-CoV-2 virus. Asterisks indicate significant differences in weight
between vehicle and MMZ treated $hACE2^{fl/y}$ animals. $N = 14$ (Vehicle) and 15 (MMZ). (**D, E**) Immunohistochemistry of WT, vehicle-treated
$hACE2^{fl/y}$, and MMZ-treated $hACE2^{fl/y}$ mouse cerebral cortex and lung 5–6 days after infection with $10^5$ PFU of SARS-CoV-2 virus using
antibodies that recognize viral nucleocapsid, the neuronal marker NeuN, the glial cell marker GFAP, the alveolar type II cell marker
DC-LAMP, the AT1 cell marker PDPN, and the nuclear stain DAPI. Arrows indicate nucleocapsid staining colocalized with NeuN+ neurons
(brain) or PDPN+ AT1 cells (lung). Representative of $N = 4$ per condition. (**F, G**) Immunohistochemistry of WT and MMZ-treated $hACE2^{fl/y}$
mouse cerebral cortex and lung 14 days after infection with $10^5$ PFU of SARS-CoV-2 virus was performed as described in D and E. Note: Day
5 $hACE2^{fl/y}$ samples (middle panel) were included on the same tissue slide as a positive control. Arrows indicate nucleocapsid staining
colocalized with neurons and AT1 cells in vehicle-treated $hACE2^{fl/y}$ mice. Representative of $N = 4$ per condition. Scale bars D-G 50 μm. (**H**)
The Foxg1$^{Cre/+}$ allele drives Cre expression in the OE and neurons of the brain (shown in red). (**I, J**) Weight loss and survival of $hACE2^{fl/y}$ and

Foxg1$^{Cre/+}$; hACE2$^{fl/y}$ mice after infection with $10^5$ viral titer per mouse. $n$ = 4 and 3 mice, respectively. (**K**) Pulse oximetry measured in WT, hACE2$^{fl/y}$, and Foxg1$^{Cre/+}$; hACE2$^{fl/y}$ mice 5–6 and 12 days after exposure to SARS-CoV-2 virus, respectively. *$p$ < 0.05, **$p$ < 0.001; ****$p$ < 0.0001 by unpaired two-tailed $t$ test, one-way ANOVA with Holm–Sidak correction for multiple comparisons, or log-rank Mantel Cox test. Numerical data in corresponding S1 Metadata tab. hACE2, human ACE2; HE, hematoxylin–eosin; MMZ, methimazole; OB, olfactory bulb; OE, olfactory epithelium; PDPN, Podoplanin; PFU, plaque-forming unit; SARS-CoV-2, Severe Acute Respiratory Syndrome Coronavirus 2; WT, wild-type.

analysis of brain lysates demonstrated partial loss of hACE2 in Baf53b-Cre; hACE2$^{fl/y}$ animals compared with hACE2$^{fl/y}$ littermate controls (Fig 7D) confirming neuronal expression of hACE2 in hACE2$^{fl/y}$ mice, while also suggesting the existence of nonneuronal sources of brain hACE2. Immunostaining for SARS-CoV-2 nucleocapsid revealed evidence of viral infection in the OE, cerebral cortex, and lung in hACE2$^{fl/y}$ animals (Fig 7E–7G). However, in Baf53b-Cre; hACE2$^{fl/y}$ mice, SARS-CoV-2 infection was detected in both the OE and lung but not the brain (Fig 7E–7G). These genetic findings demonstrate that neuronal infection is cell-autonomously ACE2-dependent and required for lethality after SARS-CoV-2 infection in hACE2$^{fl/y}$ mice.

We previously observed that infection with the Omicron BA.1 SARS-CoV-2 variant failed to confer lethal disease in of hACE2$^{fl/y}$ mice, a marked contrast to the universal lethality observed following infection with the USA-WA1 SARS-CoV-2 variant (Fig 2J and 2K). The findings described above suggested that an explanation for this difference in lethality might be a difference in the ability of the two variants to confer neuronal infection. To test this hypothesis, hACE2$^{fl/y}$ mice were infected with $10^5$ PFU of Omicron BA.1 SARS-CoV-2. Omicron BA.1 SARS-CoV-2 nucleocapsid was detected in the OE 6 days postinfection at a level similar to that observed following infection with the USA-WA1 SARS-CoV-2 variant (S14A Fig). Exposure to Omicron BA.1 SARS-CoV-2 resulted in ACE2-dependent lung infection that was evident at both 2 and 6 days postinfection (DPI), persistent pulmonary immune cell infiltrates, but was not associated with hypoxemia (S14B, S14C, and S14E-S14G Fig). However, in contrast to hACE2$^{fl/y}$ mice infected with the USA-WA1 SARS-CoV-2 variant, SARS-CoV-2 nucleocapsid was not detected in the brain 6 DPI with the Omicron BA.1 variant (S14D Fig). These findings are highly concordant with those described above and support the conclusion that neuronal infection is required for lethal COVID-19 in hACE2$^{fl/y}$ mice.

## SARS-CoV-2 infection of the lung is sufficient to confer pneumonia and transient hypoxemia but is not lethal in hACE2$^{fl/y}$ mice

The above studies demonstrated that ACE2-dependent SARS-CoV-2 infection is required in the brain but not the lung for lethal respiratory failure in hACE2$^{fl/y}$ mice. However, studies of human COVID-19 have documented significant SARS-CoV-2 infection of the lung [32,42,46,47], raising the possibility that lethal respiratory failure may also arise through direct lung infection. Moreover, hACE2$^{fl/y}$ mice express hACE2 at higher levels in the brain compared to lung, which may mask the significance of primary SARS-CoV-2 lung infection. To address this possibility and rigorously test the sufficiency of lung epithelial infection for severe disease, we generated a gain-of-function model in which Cre recombinase drives cell-specific expression of hACE2 from the *Rosa26* locus ("LSL-hACE2 mice"; Fig 8A). LSL-hACE2 mice were crossed to Shh$^{Cre}$ animals to generate Shh$^{Cre}$;LSL-hACE2$^{+/0}$ mice in which hACE2 is expressed exclusively in the epithelial cells of the lung, upper respiratory tract, and gut. Immunoblotting demonstrated strong expression of hACE2 in the lysate of lungs harvested from Shh$^{Cre}$;LSL-hACE2$^{+/0}$ but not littermate LSL-hACE2$^{+/0}$ or hACE2$^{fl/y}$ mice (Fig 8B). Immunostaining using anti-hACE2 or anti-pan-ACE2 antibodies revealed robust expression of hACE2 in the pulmonary epithelium of Shh$^{Cre}$;LSL-hACE2$^{+/0}$ animals but not LSL-hACE2$^{+/0}$ littermates (Fig 8C), consistent with Cre-dependent hACE2 expression. Exposure to SARS-CoV-2

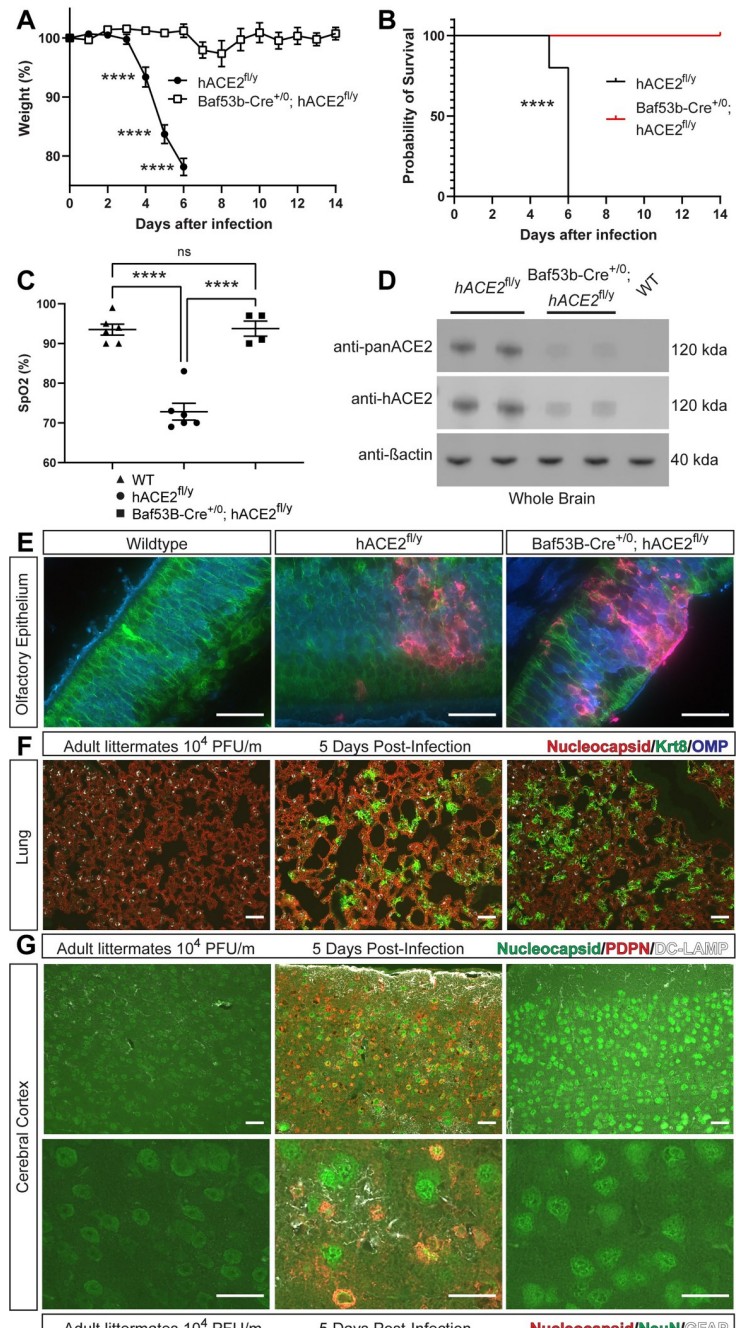

**Fig 7. Neuronal infection is required for lethal SARS-CoV-2 infection in hACE2$^{fl}$ mice. (A, B)** Weight loss and survival of $hACE2^{fl/y}$ and Baf53b-Cre; $hACE2^{fl/y}$ mice after infection with $10^4$ PFU of SARS-CoV-2. $N = 10$ ($hACE2^{fl/y}$) and 12 (Baf53b-Cre; $hACE2^{fl/y}$), three independent experiments. **(C)** Pulse oximetry measured in WT, $hACE2^{fl/y}$, and Baf53b-Cre; $hACE2^{fl/y}$ mice 6 days after exposure to $10^4$ PFU of SARS-CoV-2 virus. **(D)** Immunoblotting of whole brain lysates from $hACE2^{fl/y}$ and Baf53b-Cre; $hACE2^{fl/y}$ and WT mice was performed using anti-panACE2, anti-hACE2, and anti-β-actin antibodies. Each lane represents a single animal. $n = 4$, two independent experiments. **(E)** Immunohistochemistry of SARS-CoV-2 nucleocapsid, epithelial cell Krt8, and OSN OMP in the OE of WT, $hACE2^{fl/y}$, and Baf53b-Cre; $hACE2^{fl/y}$ mice 6 DPI. Representative of $n = 4$ animals per genotype. **(F)** Immunohistochemistry of SARS-CoV-2 nucleocapsid, AT1 cell PDPN, and AT2 cell DC-LAMP in the lungs of WT, $hACE2^{fl/y}$, and Baf53b-Cre; $hACE2^{fl/y}$ mice 6 DPI. **(G)** Immunohistochemistry of SARS-CoV-2 nucleocapsid, neuronal NeuN, and glial cell GFAP in the cerebral cortex at low and high magnification of the same tissue section of WT, $hACE2^{fl/y}$, and Baf53b-Cre; $hACE2^{fl/y}$ mice 6 DPI. Representative of $N = 5$–6 animals per genotype and time point. Scale bars in all images 50 μm. $^{****}p < 0.0001$ by unpaired two-tailed $t$ test, one-way ANOVA with Holm–Sidak correction for multiple comparisons,

or log-rank Mantel Cox test. Numerical data in corresponding S1 Metadata tab. AT1, alveolar type 1; AT2, alveolar type 2; DPI, days postinfection; OE, olfactory epithelium; OMP, olfactory marker protein; OSN, olfactory sensory neuron; PDPN, Podoplanin; PFU, plaque-forming unit; SARS-CoV-2, Severe Acute Respiratory Syndrome Coronavirus 2; WT, wild-type.

resulted in lung infection of Shh$^{Cre}$;LSL-hACE2$^{+/0}$ mice, with greater nucleocapsid staining 2 DPI than 6 DPI (Fig 8D and 8E). In contrast, SARS-CoV-2 infection was not detected in the brain of Shh$^{Cre}$;LSL-hACE2$^{+/0}$ mice (Fig 8F). HE staining of the lungs of SARS-CoV-2-infected Shh$^{Cre}$;LSL-hACE2$^{+/0}$ mice revealed the presence of bronchovascular inflammatory infiltrates and acute emphysematous changes at 2 DPI (Fig 8G). By 6 DPI, the presence of alveolar infiltrates and hyalinosis was evident (Fig 8H). This isolated SARS-CoV-2 pneumonia was associated with transient hypoxemia 10 DPI that was accompanied by a small and statistically insignificant drop in weight (Fig 8I and 8J). However, in contrast to infection of $hACE2^{fl/y}$ mice, infection of Shh$^{Cre}$;LSL-hACE2$^{+/0}$ animals did not result in prolonged weight loss or death (Fig 8J and 8K). These findings indicate that isolated SARS-CoV-2 lung infection may confer pneumonia and transient hypoxemia but is not sufficient to cause death in otherwise healthy $hACE2^{fl/y}$ mice, a finding remarkably consistent with recent studies of K18-hACE2 animals demonstrating that infection primarily in the lung is not associated with lethality [57].

## SARS-CoV-2 infection of the olfactory epithelium and/or brain is sufficient to confer lethal COVID-19 in mice

The above studies demonstrated that ACE2-dependent SARS-CoV-2 infection is required in the OE and brain but not the lung for lethal respiratory failure in $hACE2^{fl/y}$ mice and, conversely, that isolated pulmonary infection is not sufficient to cause death. These findings strongly implicated infection of nasal cavity OE and brain neurons as drivers of ARDS and lethal COVID-19 in $hACE2^{fl/y}$ mice. To test the sufficiency of OE and neuronal infection for severe disease, we generated Foxg1$^{Cre/+}$; LSL-hACE2$^{+/0}$ mice in which hACE2 is expressed exclusively in the RE, OE, and neuronal cells. Immunostaining using anti-hACE2 or anti-pan-ACE2 antibodies revealed expression of hACE2 in the OE and brain of Foxg1$^{Cre/+}$; LSL-hACE2$^{+/0}$ animals but not LSL-hACE2$^{+/0}$ littermates (S15A and S15B Fig), consistent with Cre-dependent hACE2 expression. In contrast, hACE2 could not be detected in the lung of either Foxg1$^{Cre/+}$;LSL-hACE2$^{+/0}$ or control LSL-hACE2$^{+/0}$ littermates (S15C Fig), consistent with lineage tracing studies performed by us (S10 Fig) and others [55]. Exposure of Foxg1$^{Cre/+}$; LSL-hACE2$^{+/0}$ and LSL-hACE2$^{+/0}$ mice to SARS-CoV-2 resulted in viral infection of the OE and brain but not the lung (Fig 9A–9C). However, infected Foxg1$^{Cre/+}$;LSL-hACE2$^{+/0}$ animals developed pulmonary infiltrates, intravascular thrombi, and molecular inflammatory changes identical to those observed in infected $hACE2^{fl/y}$ and Shh$^{Cre}$; $hACE2^{fl/y}$ animals (Fig 9D–9F). Finally, SARS-CoV-2 infection conferred weight loss, hypoxemia, and death in Foxg1$^{Cre/+}$; LSL-hACE2$^{+/0}$ animals, with no evidence of illness in LSL-hACE2$^{+/0}$ littermates (Fig 9G–9I).

To further distinguish between the roles and contributions of OE versus neuronal infection, we next generated Baf53b-Cre; LSL-hACE2$^{+/0}$ mice in which hACE2 is expressed exclusively in neuronal cells. Baf53b-Cre; LSL-hACE2$^{+/0}$ mice demonstrated hACE2 expression exclusively in OMP-positive OSNs and not in Keratin-8 expressing epithelial cells in the OE and in brain neurons, with no detectable expression in the lung (S16A–S16C Fig). Exposure of Baf53b-Cre; LSL-hACE2$^{+/0}$ mice to SARS-CoV-2 resulted in respiratory distress, weight loss, and sudden death at a faster tempo than that observed in $hACE2^{fl/y}$ mice (S16D and S16E Fig). Consistent with this rapid deterioration, viral nucleocapsid was detected in the OE and brain of Baf53b-Cre; LSL-hACE2$^{+/0}$ mice 2 days after infection, a time point prior to detection of

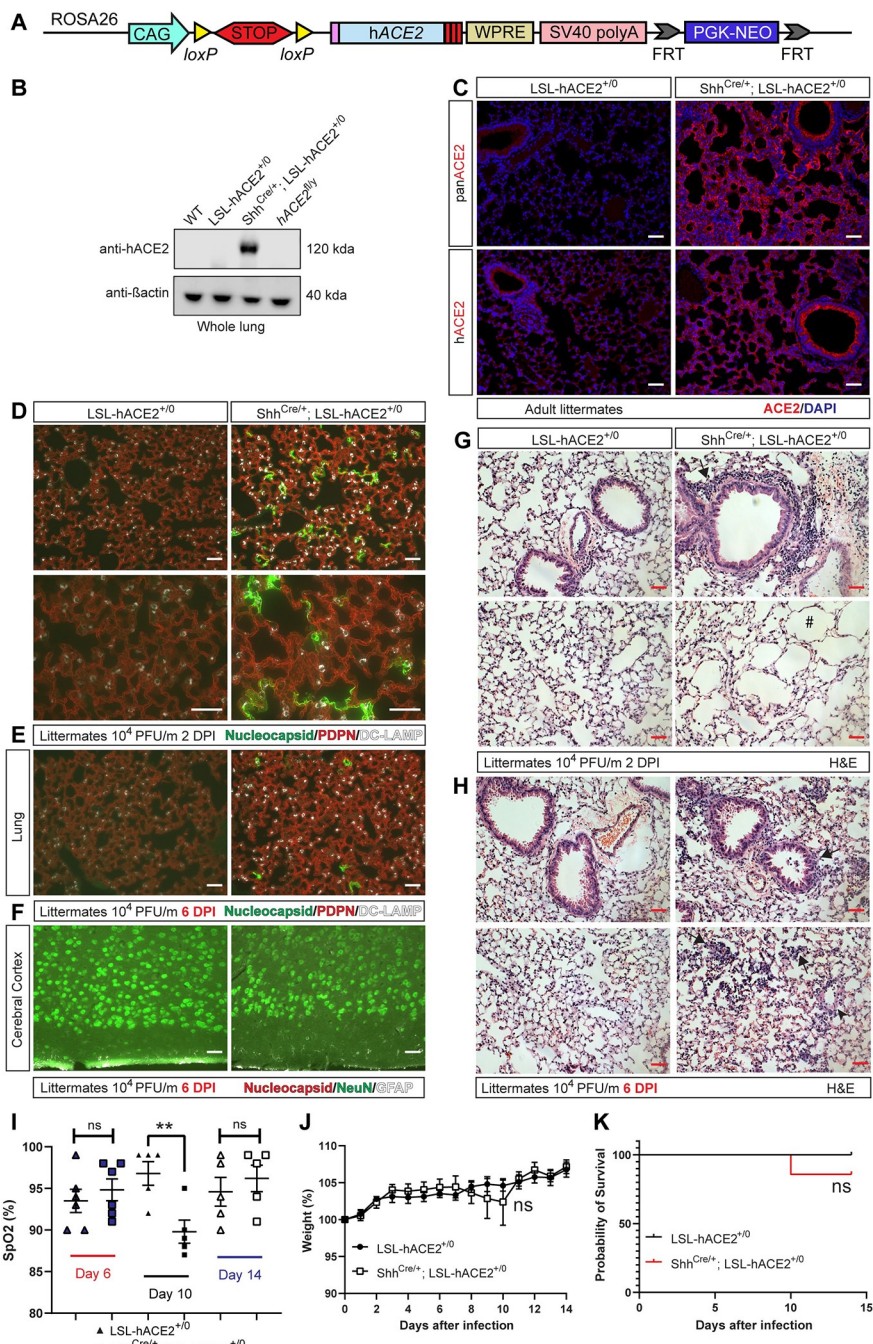

**Fig 8. Selective hACE2 expression in lung epithelial cells confers nonlethal pneumonia after SARS-CoV-2 infection.** (**A**) Generation of LSL-hACE2 mice using gene targeting of the mouse *Rosa26* locus. WPRE, SV40, PGK-NeoR. (**B**) Immunoblotting of whole lung lysates from WT, LSL-hACE2$^{+/0}$, and Shh$^{Cre/+}$;LSL-hACE2$^{+/0}$ mice was performed using anti-hACE2 and anti-β-actin antibodies. Each lane represents a single animal. Representative of $n$ = 3 per genotype and two independent experiments. (**C**) Immunohistochemistry of lung from LSL-hACE2$^{+/0}$ and Shh$^{Cre/+}$;LSL-hACE2$^{+/0}$ mice was performed using anti-panACE2 and anti-hACE2 antibodies. Representative of $N$ = 3 per genotype and two independent experiments. (**D**) Immunohistochemistry of lung from LSL-hACE2$^{+/0}$ and Shh$^{Cre/+}$;LSL-hACE2$^{+/0}$ mice was performed using antibodies to detect SARS-CoV-2 nucleocapsid, AT1 cell PDPN, and AT2 cell DC-LAMP 2 days after infection with 10$^4$ PFU of SARS-CoV-2 virus. Representative of $n$ = 4 per genotype. (**E**) Immunohistochemistry of lung from LSL-hACE2$^{+/0}$ and Shh$^{Cre/+}$;LSL-hACE2$^{+/0}$ mice was performed using antibodies to detect SARS-CoV-2 nucleocapsid, AT1 cell PDPN, and AT2 cell DC-LAMP 6 days after infection with 10$^4$ PFU of SARS-CoV-2 virus. Representative of $n$ = 4 per genotype. (**F**) Immunohistochemistry of cerebral cortex from LSL-hACE2$^{+/0}$ and Shh$^{Cre/+}$;LSL-hACE2$^{+/0}$ mice was performed using antibodies to detect SARS-CoV-2 nucleocapsid,

neuronal NeuN, and glial cell GFAP 6 days after infection with $10^4$ PFU of SARS-CoV-2 virus. Representative of $n = 4$ per genotype. (**G**) HE staining of LSL-hACE2$^{+/0}$ and Shh$^{Cre/+}$;LSL-hACE2$^{+/0}$ lung tissue 2 days after exposure to $10^4$ PFU of SARS-CoV-2 virus. Arrows, sites of inflammatory cell infiltrate. Hashtag, acute emphysematous changes. Representative of $n = 4$ animals per genotype. (**H**) HE staining of LSL-hACE2$^{+/0}$ and Shh$^{Cre/+}$;LSL-hACE2$^{+/0}$ lung tissue 6 days after exposure to $10^4$ PFU of SARS-CoV-2 virus. Arrows, sites of alveolar inflammatory infiltrate and hyalinosis. Representative of $n = 4$ animals per genotype. (**I**) Pulse oximetry measured in LSL-hACE2$^{+/0}$ and Shh$^{Cre/+}$; LSL-hACE2$^{+/0}$ mice 6, 10, and 14 days after exposure to $10^5$ PFU of SARS-CoV-2 virus. (**J, K**) Weight loss and survival of LSL-hACE2$^{+/0}$ and Shh$^{Cre/+}$;LSL-hACE2$^{+/0}$ mice after infection with $10^5$ PFU of SARS-CoV-2. $N = 6$ (LSL-hACE2$^{+/0}$) and 7 (Shh$^{Cre/+}$;LSL-hACE2$^{+/0}$ mice), two independent experiments. Scale bars in all images 50 μm. **Note:** Images in each panel were taken at lower and/or higher magnification from the same tissue section respective to genotype and highlight different pathology. ns, not significant, $p > 0.05$ **, $p < 0.01$ by unpaired two-tailed $t$ test, one-way ANOVA with Holm–Sidak correction for multiple comparisons, or log-rank Mantel Cox test. Numerical data in corresponding S1 Metadata tab. AT1, alveolar type 1; AT2, alveolar type 2; hACE2, human ACE2; HE, hematoxylin–eosin; PDPN, Podoplanin; PFU, plaque-forming unit; PGK-NeoR, Phosphoglycerate kinase promoter-driven Neomycin resistance cassette; SARS-CoV-2, Severe Acute Respiratory Syndrome Coronavirus 2; SV40, simian virus 40; WPRE, woodchuck hepatitis virus posttranscriptional regulatory element; WT, wild-type.

SARS-CoV-2 in the brains of *hACE2*$^{fl/y}$ mice (S16F and S16G Fig), with no detectable infection in the lung (S16H Fig). In contrast, isolated gain of hACE2 expression in RE in K14-Cre; LSL-hACE2$^{+/0}$ animals was insufficient to drive clinically significant disease, nor did it permit infection of the adjacent OE (S11G-S11I Fig). Taken together, these findings demonstrate that SARS-CoV-2 infection of neuronal cell populations is sufficient to confer severe COVID-19 associated with an acute lung injury, severe hypoxemia, and an ARDS phenotype in mice—findings highly consistent with the loss of function studies described above. Moreover, they demonstrate that the requirement for hACE2 remains cell autonomous even in the gain of function setting.

## Discussion

Clinical observations and analysis of postmortem human tissues have demonstrated that COVID-19 is a complex disease that affects numerous tissues, including the pulmonary, olfactory, vascular, and neurologic systems [58]. To date, a clear picture of the requisite and sufficient roles played by ACE2 and SARS-CoV-2 infection of specific cell and tissue types has yet to emerge from this large body of descriptive work. A proven approach to functional testing of disease mechanisms is genetic dissection of tissue- and cell type-specific contributions in mouse models. Our studies address COVID-19 pathogenesis using new mouse genetic models that confer lethal infection with cell-specific, Cre-mediated loss and gain of human ACE2 expression. We find that ACE2 is stringently and cell-autonomously required for SARS-CoV-2 infection of all cell types tested, including lung, olfactory epithelial cells, and neurons, in vivo. Surprisingly, our genetic dissection of ACE2 in COVID-19 reveals that lung infection by SARS-CoV-2 is neither required nor sufficient for acute lung injury, hypoxemia, and death following infection of otherwise healthy *hACE2*$^{fl/y}$ mice. Instead, genetic loss of function, genetic gain of function, and pharmacologic studies support a central role for SARS-CoV-2 infection of the OE and neuronal cells in the pathogenesis of lethal infection in mouse models of SARS-CoV-2, highlighting important considerations regarding the use of mice to study COVID-19.

During studies designed to elucidate the cause of lethality in hACE2$^{fl}$ mice, we examined the lung, nasal cavity, OE, and brain. Cre-mediated loss of hACE2 expression prevented SARS-CoV-2 infection in each organ through the cell-specific Cre activity in lung epithelium (AT1, AT2), RE, OE (sustentacular cells, OSNs), and neurons. In these organs, the infected cell types are in direct contact with each other, enabling stringent testing of the cell-specific requirement for ACE2 for SARS-CoV-2 cellular entry and infection. In all contexts, we find that ACE2 is cell-autonomously required, consistent with a primary and requisite role for

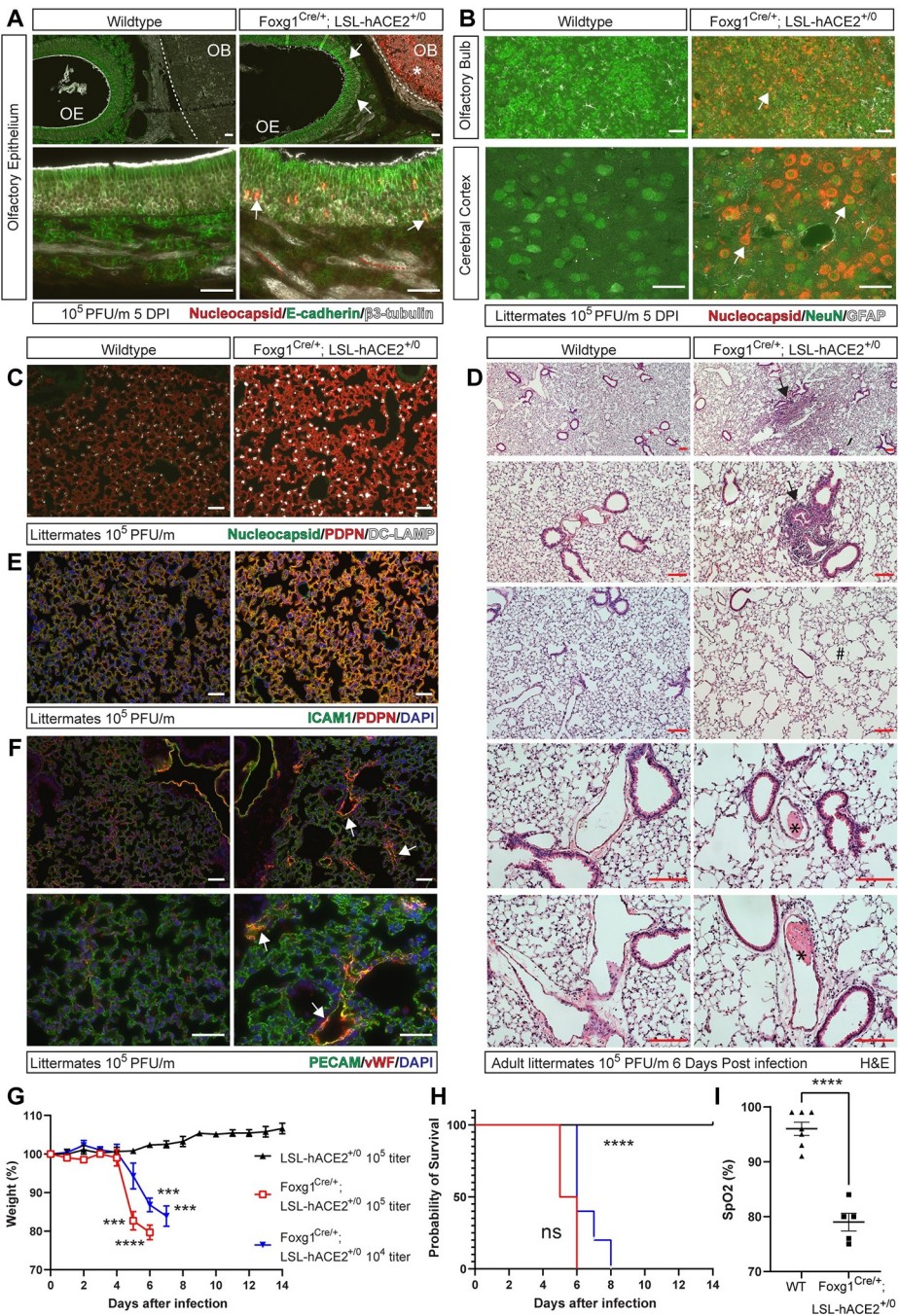

**Fig 9. Selective hACE2 expression in OE and neurons is sufficient to confer hypoxemia and death following SARS-CoV-2 infection. (A-C)** Immunohistochemistry of WT and Foxg1[Cre/+];LSL-hACE2[+/0] mouse OE (A), OB, and cerebral cortex (B) and lung (C) 6 days after infection with $10^5$ PFU of SARS-CoV-2 using antibodies that recognize viral nucleocapsid, sustentacular cell E-cadherin, olfactory sensory neuron βIII-tubulin, neuronal NeuN, glial GFAP, AT1 PDPN, or AT2 DC-LAMP. Asterisk indicates OB with nucleocapsid staining. Arrows indicate nucleocapsid staining colocalized with βIII-tubulin+ OSNs (OE in A) or NeuN+ neurons (OB and cerebral cortex in B). White dotted lines trace the border between the OE and OB. Representative of $n$ = 5 per genotype. **(D)** HE staining of WT and Foxg1[Cre/+]; $hACE2$[fl/y] lung tissue 6 days after exposure to $10^5$ PFU of SARS-CoV-2 virus. Arrows, sites of focal consolidation. Asterisks, intravascular thrombi. Representative of $n$ = 5 animals per genotype. **(E, F)** Immunohistochemistry of WT and Foxg1[Cre/+]; $hACE2$[fl/y] lung tissue 6 days after exposure to $10^5$ PFU of SARS-CoV-2 virus using antibodies against ICAM-1 and PDPN, or vWF and PECAM. Arrows in I identify vWF-positive microvasculature of the lung in Foxg1[Cre/+]; $hACE2$[fl/y] animals. Representative of $n$ = 5 animals per genotype. **(G, H)** Weight loss and survival of LSL-hACE2[+/0] and Foxg1[Cre/+]; LSL-hACE2[+/0] mice after infection with $10^5$ or $10^4$ PFU of

SARS-CoV-2. $N = 4$ (LSL-hACE2$^{+/0}$), 8 (Foxg1$^{Cre/+}$; LSL-hACE2$^{+/0}$ 10$^5$ PFU), 5 (Foxg1$^{Cre/+}$; LSL-hACE2$^{+/0}$ 10$^4$ PFU) and two independent experiments. Asterisks indicate time points at which significant differences in weight (C) or survival (D) were observed between infected Foxg1$^{Cre/+}$; LSL-hACE2$^{+/0}$ and LSL-hACE2$^{+/0}$ mice animals. (**I**) Pulse oximetry of WT (LSL-hACE2$^{+/0}$ or Foxg1$^{Cre/+}$) and Foxg1$^{Cre/+}$; LSL-hACE2$^{+/0}$ mice infected with 10$^4$ viral titer at the time of harvest (Day 6–8). ***$p < 0.001$; ****$p < 0.0001$ determined by unpaired, two-tailed $t$ test or log-rank Mantel Cox test. Scale bars in all images 50 µm. Numerical data in corresponding S1 Metadata tab. hACE2, human hACE2; ICAM-1, intracellular adhesion marker 1; OB, olfactory bulb; OE, olfactory epithelium; OSN, olfactory sensory neuron; PDPN, Podoplanin; PECAM, platelet endothelial cell adhesion molecule; PFU, plaque-forming unit; SARS-CoV-2, Severe Acute Respiratory Syndrome Coronavirus 2; vWF, von Willebrand's Factor; WT, wild-type.

membrane-bound ACE2 during viral entry in vivo. These findings suggest it is unlikely that soluble ACE2 enables infection of ACE2-deficient cells and argue against a model in which there is viral spread to cell types that do not express endogenous, membrane-bound ACE2 [8]. The stringent requirement for cellular ACE2 in viral infection and the efficiency of this molecular mechanism of pathogenesis are highlighted by our genetic findings demonstrating that ACE2 can mediate SARS-CoV-2 infection of lung AT1 cells despite expression levels that are not detectable by immunostaining. Additionally, these findings highlight the importance of functional in vivo testing using such approaches and demonstrate that descriptive approaches relying on detection of ACE2 in human tissues may not accurately define viral tropism in vivo. While our studies do not directly test or exclude a role for alternative host cell-surface receptors during viral entry and infection, they establish that ACE2 is both sufficient and required in many cell types. Thus, strategies to inhibit virus–ACE2 interaction are likely to be safe and effective in preventing viral infection.

Our efforts to generate a model of severe COVID-19 in otherwise healthy mice demonstrate that overexpression of ACE2 in the brain and lung confers highly reproducible lethality and respiratory failure following infection with the more virulent SARS-CoV-2 WA-1 strain. Whether variations in ACE2 expression contribute to disease severity in humans remains uncertain, although it is intriguing to note that genetic polymorphisms in LZTFL1 (a ciliopathy BBsome protein) and SLC6A20 (an ACE2 coreceptor) are strongly associated with severe COVID-19 [59–63] and might influence the cell-surface, ciliary expression of ACE2. Coupling these models to cell-specific loss and gain of function has proven valuable to stringently test cell-specific roles for ACE2 in vivo as we have been able to assess distinct cellular contributions to clinically relevant phenotypes in addition to descriptive studies of cellular viral infection.

On the other hand, the very strengths of our new alleles raise an obvious criticism: the possibility that nonendogenous expression of hACE2 results in artifactual pathophysiology irrelevant to human COVID-19. First, the increased level of ACE2 expression does not affect the main conclusions of this manuscript regarding a cell-autonomous requirement for ACE2 in SARS-CoV-2 infection. Second, we were struck by the potent extra-CNS effects of SARS-CoV-2 encephalitis and performed additional experiments to interrogate the relevance of this mouse pathology. While forced overexpression of ACE2 in OE and forebrain neurons (Foxg1$^{Cre}$; LSL-hACE2$^{+/0}$ and Baf53B$^{Cre}$; LSL-hACE2$^{+/0}$ in Figs 9 and S16) is sufficient to cause rapidly lethal hypoxemic respiratory failure with intravascular thromboses (hallmarks of COVID-19 acute respiratory failure), forced expression of ACE2 in the entire lung epithelium only resulted in a mild, transient hypoxemia (Shh$^{Cre}$; LSL-hACE2$^{+/0}$ in Fig 8), suggesting organ-specific responses to SARS-CoV-2 infection. Moreover, we performed a natural experiment by assessing hACE2$^{fl/y}$ mice infected with Omicron, a SARS-CoV-2 strain notable for lower human morbidity and mortality including lower incidence of neurologic symptoms (e.g., anosmia), and found that hACE2$^{fl/y}$ mice no longer exhibited neuronal infection or lethality despite the continued infection of the OE and lung epithelium. This result, where our

mouse models reflect the human COVID-19 experience, suggests that an element of host–viral interaction is independent of hACE2 expression levels.

The most severe COVID-19 presentation is one in which patients develop rapidly progressive hypoxemia associated with alveolar infiltrates and ARDS [48,49]. While respiratory failure associated with ARDS can arise due to extrapulmonary etiologies, most are associated with primary pulmonary infection and lung injury [59]. Therefore, primary SARS-CoV-2 infection of the lung has been assumed to be a causal event for respiratory failure in this disease. However, this proposed pathogenesis does not address the unique dangerousness of SARS-CoV-2 compared to other respiratory viruses, nor does it address the distinctive features of severe COVID-19 pneumonia, which necessitated changes in the clinical management of associated hypoxemic respiratory failure [64–66]. Our findings, in concert with other studies in complementary disease models along with human autopsies, raise the possibility that neuronal infection may contribute to severe COVID-19 [58]. Brain and neuronal infection, as well as acute neurologic symptoms, have been documented in COVID-19 patients and human brain organoids [67–71], but this is an inconsistent finding [52,72]. Thus, the neuronal tropism of SARS-CoV-2 and its contribution to COVID-19 pathology remains controversial.

What is the value of models in the study of human disease? Our findings contextualize many of the strengths and limitations of mouse models for investigation of COVID-19. Our studies of $hACE2^{\text{hypo/y}}$ and $hACE2^{\text{Wentworth/y}}$ mice in which hACE2 is expressed at endogenous levels in the lung and brain demonstrate that exposure of healthy animals to even high titers of SARS-CoV-2 does not confer significant disease or lethality. These findings are consistent with the fact that the vast majority of healthy, younger individuals do not exhibit severe disease after exposure to SARS-CoV-2, indicating that such models accurately reflect the human experience. However, such models are of minimal investigational value because they do not enable study of severe and complex COVID-19 symptoms that remain poorly understood at the mechanistic level. In contrast, $hACE2^{\text{fl/y}}$ and K18-hACE2 mice that express hACE2 at higher than endogenous levels in both the lung and brain exhibit reproducible respiratory failure and lethality like that experienced by very ill human patients. Similarly, wild-type mice infected with a SARS-CoV-2 virus in which a single amino acid change mutation in the spike protein (D614G) enables efficient infection and transmission do not exhibit severe disease [73], but severe disease and lethality is conferred by a more heavily mutated virus generated by multiple rounds of in vivo selection for lung infectability in mice [74,75]. These studies reveal that the most faithful mouse model is one that cannot be used to efficiently study severe COVID-19 pathogenesis in the laboratory, while those that can be used to study severe disease are able to do so because of one or another molecular bias. Therefore, how investigators use and interpret these different mouse models must take into consideration the constraints of the specific model as well as the clinical context. Despite these limitations, we believe that significant pathogenic insights will emerge from such experiments because they overcome important limitations of human COVID-19 studies that remain entirely descriptive, lack temporal resolution critical to the study of infectious disease, and are restricted to the analysis of a limited number of cells that can be harvested premortem.

## Materials and methods

### Mice

The humanized ACE2 (hACE2) floxed alleles were generated using CRISPR/Cas9-assisted mouse embryonic stem cell targeting as previously described [76]. A gRNA sequence 5′-GGATGGGATCTTGGCGCACG-3′ targeting intronic sequence immediately prior to Exon 2 of *Ace2* was cloned into the eSpCas9(1.1) plasmid (Addgene 71814). A targeting construct was

designed to insert by homologous recombination the human *ACE2* cDNA followed by a floxed WPRE-SV40 polyA and FRT-ed neomycin resistance cassette directly after the ATG-start codon of mouse *Ace2* in exon 2. The hACE2 cDNA sequence was modified to utilize the mouse ACE2 signal peptide, codon optimized for mouse codon usage, then further altered with silent mutations to remove cryptic RNA splice sites. The same hACE2 targeting construct was utilized to generate the 5′ loxP allele with an additional loxP site inserted in the 5′ untranslated region. Both targeting constructs were synthesized by Genscript. Properly targeted ES cell clones were confirmed using PCR screening of the 5′ and 3′ arms of homology and the entire targeted knock-in was PCR amplified for Sanger sequencing prior to microinjection into V6.5 ES cells. Chimeras were mated to B6D2F1/J (Jackson Laboratory, 100006) females to generate germline F1 mice. Genotyping primers that distinguish zygosity against the 5′ region of the knock-in are as follows: wild-type Forward 5′-ctcagtgcccaacccaagttc-3′, wild-type Reverse 5′-atgtcttggcattttcctcggt-3′, and mutant Reverse 5′-ggagctggagctttacggtga-3′ with expected band sizes of 400 bp (mutant) and 190 bp (wild-type).

The Rosa26-LSL-hACE2 allele was also generated using CRISPR/Cas9-assisted mouse embryonic stem cell targeting. A gRNA sequence 5′-cgtgatctgcaactccagtc-3′ targeting the XbaI restriction site conventionally used for Rosa26 targeted alleles was cloned into the eSpCas9 (1.1) plasmid (Addgene 71814). The targeting construct was synthesized by Genscript, and screening of targeted ES cell clones was performed as aforementioned. Genotyping primers against the 5′ region of the knock-in that distinguish zygosity are as follows: wild-type Forward 5′- ttctgggagttctctgctgc-3′, wild-type Reverse 5′- tgggaagtcttgtccctcca-3′, and mutant Reverse 5′- agagtgaagcagaacgtgggg-3′ with expected band sizes of 423 bp (mutant) and 210 bp (wild-type).

Shh-Cre (stock 005622), Sox2-Cre (stock 008454), Sftpc-CreERT2 (stock 028054), Hopx-CreERT2 (stock 017606), Baf53B-Cre (stock 027826); R26-LSL-tdTomato (Ai14, stock 007914) animals were purchased from Jackson Laboratories on the congenic C57BL/6J strain [45,77,78]. Shh-Cre was inherited from the male in all experiments given the known mosaic expression of Shh-Cre from maternal inheritance. Sox2-Cre was used as a germline deleter via maternal inheritance to establish a germline recombined allele. The Foxg1-IRES-Cre was a kind gift from Nada Jabado [55].

Experimental mice were maintained on a mixed C57BL/6J, 129S1/SvJ, and DBA/2J background and utilized between 2 and 4 months of age for infection studies. Mice were housed in a specific pathogen-free facility where cages were changed on a weekly basis. Cages, bedding, food, and acidified water (pH 2.5 to 3.0) were autoclaved prior to use. Ambient temperature maintained at 23°C, and 5% Clidox-S was utilized as disinfectant. The University of Pennsylvania Institutional Animal Care and Use Committee (IACUC) approved all animal protocols, and all procedures were performed in accordance with these protocols (#806811).

## Penn ABSL3 facility and viral inoculation

Animal studies were carried out in accordance with the recommendations in the Guide for the Care and Use of Laboratory Animals of the National Institutes of Health. The protocols were approved by the IACUC at the University of Pennsylvania (protocol #807017). Virus inoculations were performed under anesthesia that was induced and maintained with ketamine hydrochloride and xylazine, and all efforts were made to minimize animal suffering. Animals were housed in groups and fed standard chow diets. Mice of different ages and both sexes were administered $1 \times 10^5$ PFUs of SARS-CoV-2 via intranasal administration.

### Penn SARS-CoV-2 virus

SARS-CoV-2 (Isolate USA-WA1/2020) was obtained from BEI Resources. It was deposited by the Centers for Disease Control and Prevention and obtained through BEI Resources, NIAID, NIH: SARS-Related Coronavirus 2, Isolate USA-WA1/2020, NR-52281. Infectious stocks were grown in Vero-E6 cells and stored at −80˚C. All work with infectious virus was performed in a Biosafety Level 3 (BSL3) laboratory and approved by the Institutional Biosafety Committee and Environmental Health and Safety.

SARS-CoV-2 (Omicron) was provided by A. Pekosz (Johns Hopkins University School of Public Health). Stock virus was prepared by infection and titering of Vero E6 cells overexpressing TMPRSS2 and ACE2 and stored at −80˚C. All work with infectious virus was performed in a BSL3 laboratory and approved by the Institutional Biosafety Committee and Environmental Health and Safety.

### Cornell ABSL3 facility and viral inoculations

Mice were anesthetized with isoflurane then intranasally infected with SARS-CoV-2 USA-WA1/2020 (BEI resources; NR-52281) by applying drops of precharacterized viral stock into the rostral meatus of the nose for a total volume of 50 μL per mouse. Mice were monitored and weighed daily, then euthanized when they lost 20% of their starting weights as a predefined humane endpoint. All mice studies were performed in a BSL-3 laboratory with accordance to protocols approved by the IACUC at Cornell University (IACUC mouse protocol # 2017–0108 and BSL3 IBC # MUA-16371-1).

### Viral titers and plaque-forming unit (PFU) assays

Vero E6 cells were seeded in 12-well plates with complete DMEM and incubated at 37˚C for 24 hours prior to infection. Lung tissue was homogenized in DMEM with 2% serum at 20 μl/mg lung tissue. Supernatants were serially diluted and 100 μl used to infect cells in duplicates for 1 hour at 37˚C, gently mixed every 10 minutes, and overlayed with 1 ml of overlay medium (DMEM with 4% serum and 0.3% oxoid agar). The cells were incubated at 37˚C for 3 days, fixed with 4% PFA for 30 minutes, and then stained with 200 μl of 0.5% crystal violet in 30% methanol for 15 minutes. Crystal violet was removed, the cells were washed three times with 1 ml water, and plaques counted manually.

### Pulse oximetry

MouseStat Jr X (Kent Scientific) was used to measure heart rate and oxygen saturation (SpO2) of mice prior to harvest. To acclimate the mice to recordings, the sensor was attached daily during routine observation and weighing. Briefly, the sensor was applied to the rear leg, and data were recorded following readout stabilization defined as an unchanging recording over 5 seconds.

### Tamoxifen administration

To induce CreERT2-mediated recombination, animals at 8 weeks of age were gavaged with 100 μL of a 30-mg/mL solution of tamoxifen/corn oil (Sigma Aldrich T5648). This was done 3 consecutive days, then animals were allowed to rest for 4 days before another 3 consecutive daily doses (6 doses over 10 days). Animals were used for viral infection studies 2 weeks after completion of the tamoxifen dosing.

## Methimazole administration

A 20-mg/mL solution of MMZ/sterile PBS (Sigma Aldrich M8506) was used to administer a 100-mg/kg dose via intraperitoneal injection. Animals were inoculated with SARS-CoV-2 virus 24 to 28 hours after MMZ injection.

## Mouse harvest and tissue processing

Mice were euthanized with ketamine/xylazine and underwent laparotomy and sternotomy with subsequent left and right ventricular cardiac perfusion with 20 mL total of PBS. Trachea was exposed and cannulated to allow for bilateral lung inflation. Lungs were gently inflated with PBS infusion, which was stopped upon observation of terminal lung inflation to avoid overinflation artifacts. Right upper and middle lobes were frozen at −80˚C for downstream PFU assays. Right lower lobe was homogenized in Trizol and stored at −80˚C. Left lobe was fixed in 4% paraformaldehyde/PBS (volume/volume) along with other organs of interest for a minimum of 72 hours to ensure viral inactivation. At this point, tissues were removed from the animal BSL3 facility and underwent ethanol dehydration and embedding in paraffin blocks for histology.

The intact skull with skin/fur and eyes removed was fixed for a minimum of 72 hours and removed from the animal BSL3 facility. The lower jaw was removed, and nasal cavity was bisected in the sagittal orientation to preserve RE and OE. The nasal cavity underwent decalcification for 14 days in 0.5 M EDTA (pH 8.0) at 4˚C with agitation prior to ethanol dehydration and paraffin embedding. The brain was removed from the skull and dehydrated with ethanol for subsequent paraffin embedding and histology.

## qPCR for viral loads

Lung tissue was homogenized in Trizol using a bead mill then stored at −80˚C. Whole blood was mixed with twice the volume of Trizol LS vortexed then stored at −80˚C. RNA was extracted using chloroform phase separation and the Rneasy Mini Kit (Qiagen 74004). Approximately 1 µg of RNA was reverse transcribed with Superscript IV VILO master mix (Thermo Fisher 11756050) and diluted 1:20 prior to use in qPCR reactions. qPCR was performed using the SARS-CoV-2 Research Use Only qPCR Probe Kit (IDT 10006713) with a standard curve using a positive control N-protein plasmid (IDT 10006625).

## Immunohistochemistry

Paraffin slides with 5 µm thick tissue sections were deparaffinized with xylene and ethanol. Antigen retrieval performed with sodium citrate buffer (Sigma-Aldrich C9999) and a steamer. Primary antibodies were incubated overnight at 4˚C, and Alexa Fluor secondary antibodies were incubated at room temperature for 2 hours prior to tissue mounting with Prolong Gold (Thermo Fisher P36930). Importantly, all control and experimental samples were embedded on the same slide and underwent staining under identical conditions. Imaging was performed using identical settings.

Primary antibodies: pan-ACE2 (goat, 1:1,000, R&D AF933), hACE2 (rabbit, 1:200, Abcam ab108209), E-cadherin (rabbit, 1:200, Cell Signaling 3195S), E-cadherin (goat, 1:200, R&D AF748), olfactory marker protein (goat, 1:200, Wako 544–10001), SARS-CoV-2 nucleocapsid (rabbit, 1:500, Rockland 200-401-A50), PDPN (hamster, 1:500, Novus Biologicals AB15858), DC-LAMP (rat, 1:25, Novus Biologicals, DDX0191P-100), ICAM-1 (rabbit, 1:500, Abcam ab179707), vWF (rabbit 1:1,000, Novus Biologicals NB600-586), Endomucin (rat, 1:100, Abcam ab106100), β3-Tubulin (mouse, 1:1,000, Abcam ab78078), NeuN (mouse, 1:1,000,

Novus Biologicals NBP1-92693), GFAP (chicken, 1:1,000, Novus Biologicals NBP1-05198), Krt8 (rat, 1:100, DSHB, TROMA-I).

## Western blotting

Animal was perfused with PBS and the tissue of interest was dissected. Tissue (approximately 5 mg) was homogenized in 350 μl RIPA buffer supplemented with Protease Inhibitor Cocktail (Roche). Lysates were centrifuged at 12,000$g$ for 15 minutes at 4˚C. The supernatant was collected for downstream analysis and flash frozen prior to storage at −80˚C. Protein concentration was determined using the Pierce BCA Protein Assay Kit. Protein (30 μg) was electrophoresed on a NuPAGE 4% to 12% gel and then transferred to a nitrocellulose membrane by the iBlot2 system (Thermo Fisher Scientific) then blocked in 5% nonfat dry milk-TBST for 1 hour. Primary antibodies (diluted in 5% milk-TBST) were incubated at 4˚C overnight with gentle agitation, and membranes were then washed three times (5 minutes each) in TBST. Fluorescent-conjugated secondary antibodies (diluted in 5% milk-TBST) were applied at room temperature for 1 hour with gentle agitation, and membranes were then washed five times (5 minutes each) in TBST. Secondary antibodies were detected using the Licor Imager. Total protein counterstain was used for loading analysis (Revert, Licor 926–11011).

Primary antibodies used for immunoblotting: pan-ACE2 (1:1,000; R&D Systems; AF933), hACE2 (1:1,000; Atlas Antibodies; AMAB91259), β-Actin (1:1,000; Cell Signaling; 4970).

## RNA in situ hybridization

RNAscope in situ hybridization (ISH) was performed on decalcified paraffin tissue sections with RNAscope fluorescent multiplex reagent kit (ACD, 320850) according to the manufacturer's protocol. Briefly, sections were baked at 60˚C for 30 minutes, deparaffinized, treated with hydrogen peroxide, followed by antigen retrieval in target retrieval buffer for 15 minutes, and protease treatment before incubation with RNAscope probe for 2 hours at 40˚C. After probe hybridization, RNA signal was amplified with Amp1, Amp2, and Amp3, followed by HRP-C3 incubation. Color development was performed with TSA plus Cyanine 3 kit (PerkinElmer, NEL744B001KT). The following RNAscope probe was used: nCoV2019-S (ACD, 848561-C3).

For dual ISH and IHC, immunofluorescence staining was carried out after RNAscope ISH according to the manufacturer's protocol (ACD). Briefly, after ISH, sections were washed with PBS-T (0.1% Tween-20), blocked with 10% normal donkey serum at room temperature for 1 hour. Sections were then incubated with anti-NFH (chicken 1:5,000, Aves Labs, NFH) and anti-KRT8 (rabbit 1:500, Sigma-Aldrich, SAB4501654) overnight at 4˚C, followed by incubation with secondary antibodies.

## Antibody validation

ACE2 antibodies were validated for IHC and western blotting using hACE2$^{del/y}$ tissues. Nucleocapsid and RNAscope detection of SARS-CoV-2 was performed simultaneously on wild-type mouse tissue also infected with virus contemporaneously to experimental animals with tissue sections on the same slide to ensure identical staining conditions.

## Statistics

Animals were inoculated with SARS-CoV-2 in blinded fashion without knowledge of genotypes. At the time of harvest, hACE2 animals could not be handled in a blinded manner given moribund appearance compared to wild-type. All animals were included in analyses except those that spontaneously died prior to harvest where histology could no longer be obtained or

animals where the initial viral inoculation was observed to be inconsistent or concern for failure. Unless specifically noted, all animals used in this study were male (ACE2 is X-linked) given the excessive number of animals that would need to be generated to achieve homozygous females with additional Cre drivers. Foxg1$^{Cre}$; LSL-hACE2$^{+/0}$ experiments in Figs 9 and S10 were done with equal numbers of male and female mice. Prior determination of sample sizes could not be performed given the completely unknown phenotype of our novel genetic models. Sample sizes were predetermined to be at least three animals per genotype, which was sufficient to achieve statistical significance given the high reproducibility of our models and large effect size. Reproducibility of our findings was rigorously confirmed by infecting our novel genetic models at two separate ABSL-3 facilities with independent experimenters. Randomization of animals was not performed, favoring use of littermates to control for the mixed strain background and given the complexity of the genetic intercrosses. Statistical tests used to determine significance are described in the figure legends. Graph generation and statistical analyses performed with GraphPad Prism 9.2.0. All *t* tests performed were two-tailed. All one-way ANOVA tests were performed with Sidak's correction for multiple comparisons. Survival curve statistics were performed with log-rank Mantel Cox tests.

## Supporting information

**S1 Fig. Characterization of hACE2 expression in *hACE2*$^{fl/y}$, *hACE2*$^{hypo/y}$, and *hACE2*$^{del/y}$ mice. (A)** qPCR was performed on total RNA from whole lung from the indicated mice using primers specific for either the hACE2 cDNA (hACE2 #1 and hACE2 #2) or the WPRE cassette $^{**}$, $p < 0.01$ by unpaired, two-tailed t test. **(B)** Immunohistochemistry to detect ACE2 using human-specific ACE2 (hACE2) in the intestine and kidney of wild-type, *hACE2*$^{fl/y}$, and *hACE2*$^{del/y}$ mice. Costaining for the epithelial cell marker E-cadherin. Arrowheads indicate ACE2 staining at the brush border of the intestine (top) and in renal tubular epithelium (bottom). Representative of $N = 3$ per genotype. Scale bars 50 μm. Numerical data in corresponding Figure Data tab. ACE2, angiotensin-converting enzyme 2; hACE2, human ACE2; WPRE, woodchuck hepatitis virus posttranscriptional regulatory element.
(TIF)

**S2 Fig. Dose-dependent lethality of *hACE2*$^{fl/y}$ mice following exposure to SARS-CoV-2. (A-C)** Weight loss and survival of *hACE2*$^{fl/y}$ mice were measured after infection with the indicated PFU of SARS-CoV-2 virus. $N = 8$ and 11 for $10^5$ and $10^4$ shown in A and B. $N = 8$, 11, 8, 8, and 6 for $10^5$, $10^4$, $10^3$, $10^2$, and $10^1$ PFU in C. **(D, E)** Weight loss and survival of *hACE2*$^{fl/y}$ mice were measured after infection with the indicated PFU of SARS-CoV-2 virus at the Cornell or Penn ABSL3 facilities. $N = 8$ (Cornell) and 14 (Penn) from 3 independent experiments. **Note:** The data shown in panels A, B, and E are the same hACE2$^{fl/y}$ data shown in Figs 2 and 4. The Cornell data shown in panels C and D are the same hACE2$^{fl/y}$ data shown in Figs 2 and S6. $^*p < 0.05$; $^{***}p < 0.001$; determined by unpaired, two-tailed t-test or log-rank Mantel Cox test. Numerical data in corresponding S1 Metadata tab. ABSL3, Animal Biosafety Level 3; PFU, plaque-forming unit; SARS-CoV-2, Severe Acute Respiratory Syndrome Coronavirus 2.
(TIF)

**S3 Fig. SARS-CoV-2 infects AT2 and bronchiolar epithelial cells in the lungs of Hopx-$^{CreERT2}$; *hACE2*$^{fl/y}$ mice.** Immunohistochemistry of SARS-CoV-2 nucleocapsid, AT1 cell PDPN, and AT2 cell DC-LAMP in the lung 2 (top) and 6 (bottom) days after infection of wild-type, *hACE2*$^{fl/y}$, and Hopx$^{CreERT2}$; *hACE2*$^{fl/y}$ mice. Hashtags in the center of bronchi. Representative of $N = 4$–5 animals per genotype and time point. Scale bars 50 μm. AT1, alveolar type 1; AT2, alveolar type 2; PDPN, Podoplanin; SARS-CoV-2, Severe Acute Respiratory Syndrome

Coronavirus 2.
(TIF)

**S4 Fig. Shh$^{Cre}$ is active in lower RE but spares the nasal cavity epithelium. (A)** Sites of Shh$^{Cre}$ activity in respiratory and gut epithelium are shown in red. * indicates OB. Lines trace regions of nasal cavity epithelium. (**B**) Lineage trace of Shh$^{Cre}$ activity in upper and lower RE using a Cre-activated, tdTomato allele (LSL-RFP). Immunohistochemistry using E-cadherin (epithelium) and RFP (Cre reporter) antibodies is shown. $N$ = 3 per genotype. Scale bars 100 μm. **Note:** The LSL-RFP$^{+/0}$ data (left) in panel B are the same in S11A Fig. While these animals were not littermates, the respective tissue was sectioned and immunostained contemporaneously on the same slide for stringent comparison. OB, olfactory bulb; OE, olfactory epithelium; RE, respiratory epithelium; RFP, Red Fluorescent Protein; TZ, transition zone.
(TIF)

**S5 Fig. Loss of epithelial expression of hACE2 in the intestine of Shh$^{Cre}$; $hACE2^{fl/y}$ mice.** Immunohistochemistry of wild-type, $hACE2^{fl/y}$, and Shh$^{Cre}$; $hACE2^{fl/y}$ small intestine using antibodies that recognize both human and mouse ACE2 (pan-ACE2) and only hACE2 are shown. Scale bars 50 μm. Representative of $n$ = 3 per genotype and 2 independent litters. hACE2, human ACE2.
(TIF)

**S6 Fig. Acute lung injury, hypoxemia, and lethality in $hACE2^{fl/y}$ and Shh$^{Cre/+}$; $hACE2^{fl/y}$ mice after infection with SARS-CoV-2. (A, B)** Weight loss and survival of $hACE2^{fl/y}$ and Shh$^{Cre/+}$; $hACE2^{fl/y}$ mice after infection with $10^5$ PFU of SARS-CoV-2. $N$ = 8 ($hACE2^{fl/y}$) and 7 (Shh$^{Cre/+}$; $hACE2^{fl/y}$), two independent experiments. Note: Data for $hACE2^{fl/y}$ are the same as those shown in Fig 2D and 2E because Shh$^{Cre/+}$; $hACE2^{fl/y}$ animals were littermates of those animals. (**C**) Pulse oximetry measured in WT, $hACE2^{fl/y}$, and Shh$^{Cre/+}$; $hACE2^{fl/y}$ mice at time of harvest 5–6 days after exposure to $10^5$ PFU of SARS-CoV-2 virus. (**D**) HE staining of WT, $hACE2^{fl/y}$, and Shh$^{Cre/+}$; $hACE2^{fl/y}$ lung tissue 6 days after exposure to $10^5$ PFU of SARS-CoV-2 virus. Arrows, sites of bronchovascular immune cell infiltrate. Asterisks, intravascular thrombus. Representative of $N$ = 3 animals per genotype. Scale bars 100 μm. **Note:** The $hACE2^{fl/y}$ data shown in panels A and B are the same as those shown in Fig 2, panels B-G. $^*p < 0.05$, $^{**}p < 0.01$, ns $p > 0.05$ by unpaired two-tailed $t$ test, one-way ANOVA with Holm–Sidak correction for multiple comparisons, or log-rank Mantel Cox test. These $hACE2^{fl/y}$ mice were littermates of the Shh$^{Cre/+}$; $hACE2^{fl/y}$ mice and contemporaneously infected. Numerical data in corresponding S1 Metadata tab. HE, hematoxylin–eosin; PFU, plaque-forming unit; SARS-CoV-2, Severe Acute Respiratory Syndrome Coronavirus 2; WT, wild-type.
(TIF)

**S7 Fig. Extensive SARS-CoV-2 detection in hippocampus but minimal detection in neurons of the cerebellum and brainstem.** Immunohistochemistry of wild-type, $hACE2^{fl/y}$, and Shh$^{Cre/+}$; $hACE2^{fl/y}$ hippocampus, cerebellum, and brainstem with antibody staining for viral nucleocapsid, neurons (NeuN), and glial cells (GFAP) is shown 5–6 days after infection with SARS-CoV-2. Arrows indicate NeuN-positive neurons with colocalized nucleocapsid staining. $N$ = 3 per genotype. Scale bars 100 μm (top three rows), 50 μm (bottom two rows). SARS-CoV-2, Severe Acute Respiratory Syndrome Coronavirus 2.
(TIF)

**S8 Fig. Meningeal and vascular inflammation is associated with neuronal infection at the choroid plexus. (A, B)** Immunohistochemistry of SARS-CoV-2 nucleocapsid, neuronal NeuN, and glial cell GFAP of the cerebral cortex adjacent to the choroid plexus 2 and 5–6 days

after infection. Arrows indicated nucleocapsid staining. Red dotted lines trace ependymal cells of the choroid plexus. (**C, D**) HE staining of brain choroid plexus and cerebral cortex blood vessels 2 and 5–6 days after infection of *hACE2*[fl/y] and Shh[Cre/+]; *hACE2*[fl/y] mice. Arrows indicate sites of immune cell infiltration. Representative of *N* = 4 animals per genotype and time point. Scale bars 50 μm. HE, hematoxylin–eosin; SARS-CoV-2, Severe Acute Respiratory Syndrome Coronavirus 2.
(TIF)

**S9 Fig. Failure of MMZ to prevent lethal SARS-CoV-2 infection is associated with brain infection.** (**A, B**) Cerebral cortex (A) and lung (B) immunohistochemistry of *hACE2*[fl/y] animals treated with vehicle or MMZ followed by infection with $10^5$ viral titer. The tissues shown on the far right ("*hACE2*[fl] + MMZ 8 DPI") are from a *hACE2*[fl] mouse that exhibited weight loss and required euthanasia 8 days following infection with $10^5$ SARS-CoV-2 despite pretreatment with MMZ. Cerebral cortex stained with viral nucleocapsid, neuronal NeuN, and glial cell GFAP. Lung stained with viral nucleocapsid, AT1 cells PDPN, and AT2 cell DC-LAMP. Arrows indicate nucleocapsid staining colocalized with NeuN+ neurons or PDPN+ AT1 cells. *N* = 4 per genotype except for *N* = 1 (MMZ treated). Scale bars 50 μm. AT1, alveolar type 1; AT2, alveolar type 2; MMZ, methimazole; SARS-CoV-2, Severe Acute Respiratory Syndrome Coronavirus 2.
(TIF)

**S10 Fig. Foxg1**[Cre] **is active in neurons and nasal cavity epithelium but not in tracheal and lung epithelium.** Foxg1[Cre] was crossed to a Cre-activated, tdTomato allele (LSL-RFP) to trace activity along the route of SARS-CoV-2 infection. Sites of Foxg1[Cre] activity are shown in red. (**A, B**) Immunostaining to detect E-cadherin, NeuN, and RFP demonstrate that Foxg1[Cre] is active in RE, OE, and neurons in the brain. (**B**) Immunostaining to detect E-cadherin, AQP5, PDPN, and RFP demonstrate that Foxg1[Cre] is not active in tracheal or lung epithelial cells. Cre activity was detected in an E-cadherin negative, sub-epithelial population of cells in the tracheal. *N* = 3 per genotype. Scale bars 50 μm. AQP5, Aquaporin 5; OE, olfactory epithelium; PDPN, Podoplanin; RE, respiratory epithelium; RFP, Red Fluorescent Protein; SARS-CoV-2, Severe Acute Respiratory Syndrome Coronavirus 2.
(TIF)

**S11 Fig. hACE2 in nasal cavity respiratory and transition zone epithelium is not required for lethal SARS-CoV-2 infection.** (**A**) A K14-Cre transgene was crossed to a Cre-activated, tdTomato allele (LSL-RFP) to trace activity along the route of SARS-CoV-2 infection. Immunostaining for pan-epithelial cell marker E-cadherin and Cre reporter RFP was performed in nasal cavity, trachea and lung. *n* = 3. Scale bars 100 μm. (**B, C**) Survival and weight loss of *hACE2*[fl/y] and K14-Cre; *hACE2*[fl/y] mice after infection with $10^4$ PFU of SARS-CoV-2. *N* = 5 (*hACE2*[fl/y]) and 5 (K14-Cre; *hACE2*[fl/y]), one experiment. (**D**) Pulse oximetry measured in WT, *hACE2*[fl/y], and K14-Cre; *hACE2*[fl/y] mice 6 days after exposure to $10^4$ PFU of SARS-CoV-2. (**E, F**) Immunohistochemistry of WT, *hACE2*[fl/y], and K14-Cre; *hACE2*[fl/y] RE, OE, and lung tissue 2 days after exposure to $10^4$ PFU of SARS-CoV-2 was performed using antibodies against E-cadherin (epithelial cells), SARS-CoV-2 nucleocapsid, β3-tubulin (neurons), DC-LAMP (alveolar type 2 cells), and PDPN (alveolar type 1 cells). Note the loss of viral infection of RE but not OE or lung epithelium in K14-Cre; *hACE2*[fl/y] mice. Representative of *N* = 4 animals per genotype. Scale bars, 50 μm. (**G, H**) Weight loss and survival of LSL-hACE2[+/0] and K14-Cre; LSL-hACE2[+/0] mice after infection with $10^4$ PFU of SARS-CoV-2. *N* = 3 (LSL-hACE2[+/0]) and 8(K14-Cre; LSL-hACE2[+/0]). (**I**) Pulse oximetry of WT (LSL-hACE2[+/0] or K14-Cre;*Ace2*[+/+]) and K14-Cre; LSL-hACE2[+/0] mice infected with $10^4$ viral titer 6 days after

infection. **Note:** The LSL-RFP$^{+/0}$ data (left) in panel A are the same in S4B Fig. While these animals were not littermates, the respective tissue was sectioned and immunostained contemporaneously on the same slide for stringent comparison. ns, not significant $p > 0.05$. $^{**}p < 0.01$ by unpaired two-tailed $t$ test, one-way ANOVA with Holm–Sidak correction for multiple comparisons, or log-rank Mantel Cox test. Numerical data in corresponding S1 Metadata tab. hACE2, human ACE2; OE, olfactory epithelium; PFU, plaque-forming unit; RE, respiratory epithelium; RFP, Red Fluorescent Protein; SARS-CoV-2, Severe Acute Respiratory Syndrome Coronavirus 2; WT, wild-type.
(TIF)

**S12 Fig. Foxg1$^{Cre}$-mediated loss of hACE2 prevents infection of nasal epithelium but not lung epithelium in *hACE2*$^{fl/y}$ mice. (A, B)** Immunohistochemistry of SARS-CoV-2 nucleocapsid, epithelial E-cadherin, neuronal β3 tubulin, AT1 cell PDPN, and AT2 cell DC-LAMP in the respiratory, olfactory, and pulmonary epithelium 2 days after infection of wild-type, *hACE2*$^{fl/y}$, and Foxg1$^{Cre}$; *hACE2*$^{fl/y}$ mice. Representative of $N = 3$ animals per genotype and time point. Scale bars 50 μm. **(C, D)** Immunohistochemistry of SARS-CoV-2 nucleocapsid, epithelial E-cadherin, neuronal β3 tubulin, AT1 cell PDPN, and AT2 cell DC-LAMP in the respiratory, olfactory, and pulmonary epithelium 6 days after infection of wild-type, *hACE2*$^{fl/y}$, and Foxg1$^{Cre}$; *hACE2*$^{fl/y}$ mice. Representative of $N = 4$ animals per genotype and time point. Scale bars 50 μm. AT1, alveolar type 1; AT2, alveolar type 2; hACE2, human ACE2; SARS-CoV-2, Severe Acute Respiratory Syndrome Coronavirus 2.
(TIF)

**S13 Fig. Baf53b-Cre is active in neurons of the OE and brain but not in trachea or lung epithelium.** Lineage tracing of Baf53b-Cre activity along the route of SARS-CoV-2 infection was performed using a Cre-activated, tdTomato allele (LSL-RFP). **(A)** Immunostaining for RFP and OMP, a marker of OSNs, in OE LSL-hACE2$^{+/0}$ and Baf53b-Cre; LSL-hACE2$^{+/0}$ mice. **(B)** Immunostaining for RFP and NeuN, a neuronal marker, in the brain of LSL-hACE2$^{+/0}$ and Baf53b-Cre; LSL-hACE2$^{+/0}$ mice. **(C, D)** Immunostaining for RFP and E-cadherin, an epithelial cell marker, in the trachea and lung of LSL-hACE2$^{+/0}$ and Baf53b-Cre; LSL-hACE2$^{+/0}$ mice. $N = 3$ per genotype. Scale bars in all images 50 μm. OE, olfactory epithelium; OMP, olfactory marker protein; OSN, olfactory sensory neuron; RFP, Red Fluorescent Protein; SARS-CoV-2, Severe Acute Respiratory Syndrome Coronavirus 2.
(TIF)

**S14 Fig. The Omicron BA.1 SARS-CoV-2 variant infects the OE and lung but not the brain of *hACE2*$^{fl/y}$ mice. (A)** Immunostaining for SARS-CoV-2 nucleocapsid in the OE of *hACE2*$^{fl/y}$ mice 6 days after infection with the indicated doses of USA-WA1 or Omicron BA.1 variants. **(B)** Immunohistochemistry of wild-type, *hACE2*$^{fl/y}$, and Shh$^{Cre}$; *hACE2*$^{fl/y}$ lung 2 DPI with $10^5$ PFU of Omicron BA.1 SARS-CoV-2 virus was performed using antibodies that recognize viral nucleocapsid, AT1 cell PDPN, and AT2 cell DC-LAMP. $N = 4$ per genotype. **(C)** Immunohistochemistry of *hACE2*$^{fl/y}$ lung 6 DPI with $10^4$ PFU of USA-WA1 or $10^5$ PFU of Omicron BA.1 SARS-CoV-2 virus was performed using antibodies that recognize viral nucleocapsid, AT1 cell PDPN, and AT2 cell DC-LAMP. $N = 5$ per genotype. **(D)** Immunohistochemistry of *hACE2*$^{fl/y}$ lung 6 DPI with $10^4$ PFU of USA-WA1 or $10^5$ PFU of Omicron BA.1 SARS-CoV-2 virus was performed using antibodies that recognize viral nucleocapsid, neuronal NeuN, and glial cell GFAP. $N = 5$ per genotype. **(E)** Pulse oximetry of *hACE2*$^{fl/y}$ mice infected with $10^4$ PFU of USA-WA1 or $10^5$ PFU of Omicron BA.1 SARS-CoV-2 virus 6 DPI. $^{****}p < 0.0001$ determined by unpaired, two-tailed $t$ test. **(F)** HE staining of wild-type and *hACE2*$^{fl/y}$ lung tissue 6 (F) and 12 (G) days after exposure to $10^5$ PFU of SARS-CoV-2 virus. Representative of $n = 3$ animals

per genotype. Scale bars in all images 50 μm. Numerical data in corresponding S1 Metadata tab. AT1, alveolar type 1; AT2, alveolar type 2; DPI, days postinfection; OE, olfactory epithelium; PDPN, Podoplanin; PFU, plaque-forming unit; SARS-CoV-2, Severe Acute Respiratory Syndrome Coronavirus 2.
(TIF)

**S15 Fig. hACE2 expression in OE and neurons but not lung epithelium of Foxg1$^{Cre}$;LSL-hACE2$^{+/0}$ mice.** (**A-C**) Expression of hACE2 was detected using either anti-hACE2 or anti-panACE2 antibodies in Foxg1$^{Cre}$;LSL-hACE2$^{+/0}$ and LSL-hACE2$^{+/0}$ littermate control mice. Costaining for E-cadherin, OMP, NeuN, AQP5, and CD-LAMP was used to label epithelial cells, OSNs, neurons, AT1, and AT2 cells, respectively. Arrows indicate sites of hACE2 expression in OE and cerebral cortex. $N$ = 3 per genotype. Scale bars 50 μm. AQP5, Aquaporin 5; AT1, alveolar type 2; AT2, alveolar type 2; hACE2, human ACE2; OE, olfactory epithelium; OMP, olfactory marker protein; OSN, olfactory sensory neuron.
(TIF)

**S16 Fig. Selective hACE2 expression in neurons is sufficient to confer weight loss and death following SARS-CoV-2 infection.** (**A-C**) Immunohistochemistry of LSL-hACE2$^{+/0}$, Baf53b-Cre$^{+/0}$;LSL-hACE2$^{+/0}$, and hACE2$^{fl/y}$ mouse OE (A), brain (B), and lung (C) using antibodies that recognize hACE2 and markers of OE (Krt8 and OMP), neurons (NeuN), and lung epithelial cells (DC-LAMP and PDPN). Arrows indicate OSNs and arrowheads indicate OSN cilia in (A). Arrows indicate faint expression of hACE2 in DC-LAMP+ AT2 cells in (C). Representative of $n$ = 3 per genotype. (**D, E**) Weight loss and survival of LSL-hACE2$^{+/0}$, Baf53b-Cre$^{+/0}$;LSL-hACE2$^{+/0}$, and hACE2$^{fl/y}$ mice after infection with $10^5$ PFU of SARS-CoV-2. $n$ = 5 for each genotype. Asterisks indicate time points at which significant differences in weight (D) or survival (E) were observed between Baf53b-Cre$^{+/0}$;LSL-hACE2$^{+/0}$ and hACE2$^{fl/y}$ animals compared to contemporaneously infected LSL-hACE2$^{+/0}$ controls. (**F-H**) Immunohistochemistry of SARS-CoV-2 nucleocapsid and markers of OE (Krt8 and OMP), neurons (NeuN), glial cells (GFAP), and lung epithelial cells (DC-LAMP and PDPN) in the OE, brain, and lung 2 days after infection of LSL-hACE2$^{+/0}$, Baf53b-Cre$^{+/0}$;LSL-hACE2$^{+/0}$, and hACE2$^{fl/y}$ mice. Arrows indicate sites of viral nucleocapsid detection. Representative of $N$ = 3 animals per genotype and time point. Scale bars in all images 50 μm. $^*p < 0.05$; $^{**}p < 0.01$; $^{****}p < 0.0001$ determined by unpaired, two-tailed $t$ test or log-rank Mantel Cox test. Numerical data in corresponding S1 Metadata tab. AT2, alveolar type 2; hACE2, human ACE2; OE, olfactory epithelium; OMP, olfactory marker protein; OSN, olfactory sensory neuron; PDPN, Podoplanin; PFU, plaque-forming unit; SARS-CoV-2, Severe Acute Respiratory Syndrome Coronavirus 2.
(TIF)

**S1 Metadata. Underlying numerical data to graphs throughout the manuscript.** An Excel file with separate sheets for each figure panel where numerical data were graphed.
(XLSX)

**S1 Raw Images. Original western blot images.** In this file are unedited western blot images corresponding to those presented throughout the manuscript. The edited western blot is shown above the unedited blot.
(PDF)

## Acknowledgments

We thank the members of the Kahn lab for their thoughtful comments and advice during this work. Schematics of the mouse brain and respiratory tract were created with BioRender.com

and granted license to be published under Academic License Terms GC24SFS4XP, VY24S-FRI1O, RG24SFRD38, and HW24SFR8F7.

## Author Contributions

**Conceptualization:** Alan T. Tang, David W. Buchholz, Katherine M. Szigety, Brian Imbiakha, Siqi Gao, Isaac A. Monreal, Avery August, Kellie A. Jurado, Mingang Xu, Edward E. Morrisey, Sarah E. Millar, Hector C. Aguilar, Mark L. Kahn.

**Data curation:** Alan T. Tang, David W. Buchholz, Katherine M. Szigety, Brian Imbiakha, Siqi Gao, Maxwell Frankfurter, Min Wang, Peter Hewins, Julie Sahler, Xuming Zhu, Mingang Xu, Sarah E. Millar, Hector C. Aguilar, Mark L. Kahn.

**Formal analysis:** Alan T. Tang, David W. Buchholz, Katherine M. Szigety, Brian Imbiakha, Siqi Gao, Maxwell Frankfurter, Min Wang, Peter Hewins, Avery August, Xuming Zhu, Kellie A. Jurado, Mingang Xu, Sarah E. Millar, Hector C. Aguilar, Mark L. Kahn.

**Funding acquisition:** Patricia Mericko-Ishizuka, Edward E. Morrisey, Sarah E. Millar, Hector C. Aguilar, Mark L. Kahn.

**Investigation:** Alan T. Tang, Katherine M. Szigety, Min Wang, Jisheng Yang, N Adrian Leu, Julie Sahler, Mingang Xu, Edward E. Morrisey, Sarah E. Millar, Hector C. Aguilar, Mark L. Kahn.

**Methodology:** Alan T. Tang, Min Wang, Jisheng Yang, Peter Hewins, Patricia Mericko-Ishizuka, N Adrian Leu, Stephanie Sterling, Julie Sahler, Xuming Zhu, Mingang Xu, Edward E. Morrisey, Sarah E. Millar, Hector C. Aguilar, Mark L. Kahn.

**Project administration:** Patricia Mericko-Ishizuka, Hector C. Aguilar, Mark L. Kahn.

**Resources:** N Adrian Leu, Stephanie Sterling, Hector C. Aguilar.

**Writing – original draft:** Alan T. Tang, David W. Buchholz, Katherine M. Szigety, Brian Imbiakha, Siqi Gao, Maxwell Frankfurter, Min Wang, Kellie A. Jurado, Edward E. Morrisey, Sarah E. Millar, Hector C. Aguilar, Mark L. Kahn.

**Writing – review & editing:** Alan T. Tang, David W. Buchholz, Katherine M. Szigety, Edward E. Morrisey, Sarah E. Millar, Hector C. Aguilar, Mark L. Kahn.

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
