## [Editor Report · Decision Letter 0]

21 Nov 2022

Dear Dr. Kahn, 

Thank you for submitting your manuscript entitled "Cell-autonomous requirement for ACE2 across organs in lethal SARS-CoV-2 infection in mice" for consideration as a Research Article by PLOS Biology.

Your manuscript has now been evaluated by the PLOS Biology editorial staff, as well as by an academic editor with relevant expertise, and I am writing to let you know that we would like to send your submission out for external peer review.

Once your full submission is complete, your paper will undergo a series of checks in preparation for peer review. After your manuscript has passed the checks it will be sent out for review. To provide the metadata for your submission, please Login to Editorial Manager (https://www.editorialmanager.com/pbiology) within two working days, i.e. by Nov 23 2022 11:59PM.

Kind regards,

Paula

---

Senior Editor

PLOS Biology

---

## [Editor Report · Decision Letter 1]

13 Dec 2022

Dear Dr. Kahn,

Thank you for your patience while your manuscript "Cell-autonomous requirement for ACE2 across organs in lethal SARS-CoV-2 infection in mice" was being assessed at PLOS Biology. Your work, the previous reviews and your responses to reviewers have now been evaluated by the PLOS Biology editors and an Academic Editor with relevant expertise.

Based on our Academic Editor's assessment of your revision, we are likely to accept this manuscript for publication, provided you satisfactorily address the following data and other policy-related requests.

1. DATA POLICY:

A) Supplementary files (e.g., excel). Please ensure that all data files are uploaded as 'Supporting Information' and are invariably referred to (in the manuscript, figure legends, and the Description field when uploading your files) using the following format verbatim: S1 Data, S2 Data, etc. Multiple panels of a single or even several figures can be included as multiple sheets in one excel file that is saved using exactly the following convention: S1_Data.xlsx (using an underscore).

B) Deposition in a publicly available repository. Please also provide the accession code or a reviewer link so that we may view your data before publication.

Regardless of the method selected, please ensure that you provide the individual numerical values that underlie the summary data displayed in the following figure panels as they are essential for readers to assess your analysis and to reproduce it: Figures 1C, 2BCDEFGHIJK, 3CEGH, 4ABC, 6BCIJK, 7ABC, 8IJK, 9GHI, and supplementary figures S1A, S2ABCDE, S6ABC, S11BCDGHI, S14E, S16DE.

**Please also ensure that figure legends in your manuscript include information on where the underlying data can be found, and ensure your supplemental data file/s has a legend.**

We require the original, uncropped and minimally adjusted images supporting all blot and gel results reported in an article's figures or Supporting Information files. We will require these files before a manuscript can be accepted so please prepare and upload them now. We need this for figures 1BDEFG, 7D, 8B.

Please carefully read our guidelines for how to prepare and upload this data: https://journals.plos.org/plosbiology/s/figures#loc-blot-and-gel-reporting-requirements

We expect to receive your revised manuscript within two weeks.

*Published Peer Review History*

*Press*

Sincerely,

Paula

---

Senior Editor,

pjaureguionieva@plos.org,

PLOS Biology

---

## [Editor Report · Decision Letter 2]

4 Jan 2023

Dear Dr Kahn,

Thank you for the submission of your revised Research Article "Cell-autonomous requirement for ACE2 across organs in lethal mouse SARS-CoV-2 infection" for publication in PLOS Biology. On behalf of my colleagues and the Academic Editor, Frank Kirchhoff, I am pleased to say that we can in principle accept your manuscript for publication, provided you address any remaining formatting and reporting issues. These will be detailed in an email you should receive within 2-3 business days from our colleagues in the journal operations team; no action is required from you until then. Please note that we will not be able to formally accept your manuscript and schedule it for publication until you have completed any requested changes.

PRESS

Sincerely, 

Paula 

---

Senior Editor

PLOS Biology
